# Structural basis of Spliced Leader RNA recognition by the *Trypanosoma brucei* cap-binding complex

Harald Bernhard[1,6], Hana Petržílková [1], Barbora Popelářová [1,7], Kamil Ziemkiewicz[2], Karolina Bartosik [3], Marcin Warmiński [4], Laura Tengo [1], Henri Gröger [1,6], Luciano G. Dolce [1,8], Cameron D. Mackereth [5], Ronald Micura [3], Jacek Jemielity [2] & Eva Kowalinski [1] ✉

Kinetoplastids are a clade of eukaryotic protozoans that include human parasitic pathogens like trypanosomes and Leishmania species. In these organisms, protein-coding genes are transcribed as polycistronic pre-mRNAs, which need to be processed by the coupled action of trans-splicing and polyadenylation to yield monogenic mature mRNAs. During trans-splicing, a universal RNA sequence, the spliced leader RNA (SL RNA) mini-exon, is added to the 5'-end of each mRNA. The 5'-end of this mini-exon carries a hyper-methylated cap structure and is bound by a trypanosomatid-specific cap-binding complex (CBC). The function of three of the kinetoplastid CBC sub-units is unknown, but an essential role in cap-binding and trans-splicing has been suggested. Here, we report cryo-EM structures that reveal the molecular architecture of the *Trypanosoma brucei* CBC (*Tb*CBC) complex. We find that *Tb*CBC interacts with two distinct features of the SL RNA. The *Tb*CBP20 sub-unit interacts with the m7G cap while *Tb*CBP66 recognizes double-stranded portions of the SL RNA. Our findings pave the way for future research on mRNA maturation in kinetoplastids. Moreover, the observed structural similarities and differences between *Tb*CBC and the mammalian cap-binding complex will be crucial for considering the potential of *Tb*CBC as a target for anti-trypanosomatid drug development.

Kinetoplastids are flagellated eukaryotic parasitic protists. Trypano-somatids, a subgroup of kinetoplastids, are responsible for various human and animal diseases, that pose a substantial public health burden and worldwide economic challenges. Diseases caused by these parasites include different forms of Leishmaniasis caused by Leishmania parasites or trypanosomatid diseases like African trypanosomiasis, Nagana or Chagas disease caused by different *Trypanosoma brucei* sub-species or *Trypanosoma cruzi*[1–4].

Kinetoplastids transcribe their DNA as polycistronic pre-mRNA cassettes that contain clusters of genes of unrelated function

[1]EMBL Grenoble, 71 Avenue des Martyrs, Grenoble, France. [2]Centre of New Technologies, University of Warsaw, Warsaw, Poland. [3]Institute of Organic Chemistry, Center for Molecular Biosciences Innsbruck, University of Innsbruck, Innsbruck, Austria. [4]Division of Biophysics, Institute of Experimental Physics, Faculty of Physics, University of Warsaw, Warsaw, Poland. [5]University of Bordeaux, INSERM, CNRS, ARNA Laboratory, U1212, UMR 5320, Bordeaux, France. [6]Present address: Institut de Biologie Structurale (IBS), Université Grenoble Alpes (UGA), Commissariat à l'Energie Atomique et aux Energies Alternatives (CEA), Centre National de la Recherche Scientifique (CNRS), Grenoble, France. [7]Present address: Department of Experimental Biology, Section of Micro-biology, Faculty of Science, Masaryk University, Brno, Czech Republic. [8]Present address: Institute for Advanced Biosciences (IAB), INSERM U1209, CNRS UMR 5309, Université Grenoble-Alpes, Grenoble, France. ✉e-mail: kowalinski@embl.fr

(reviewed in refs. 5–7). These pre-mRNAs are processed through the concerted action of trans-splicing and polyadenylation, which generates individual capped and polyadenylated monocistronic mRNAs[7–12]. Because it is an obligatory processing step for almost all mRNAs, trans-splicing is vital for trypanosomes. During the trans-splicing process, the spliceosome catalyzes the addition of a universal 39 nt RNA mini-exon to the 5′-end of each expression unit. Trans-splicing produces a Y-structured RNA intermediate by-product analogous to the circular intron lariat formed during the cis-splicing process[13,14]. The added mini-exon originates from the approximate 140 nt spliced leader RNA (SL RNA), which is transcribed separately in large copy numbers[15]. The SL RNA adopts a secondary structure resembling the structured RNAs contained in small nuclear ribonucleoproteins (snRNPs), which are fundamental components of the eukaryotic RNA spliceosome[16,17]. Like snRNPs, SL RNA is associated with proteins to form an SL snRNP particle, but this complex is unique to the kinetoplastid trans-splicing process[18–22]. The 5′-end of the SL RNA of trypanosomatids carries a hypermethylated structure containing seven base and ribose methylations within the first four nucleotides, termed cap4 ($m^7GpppA_mpA_mpC_mp^{m3}U_mp$) (Fig. 1A)[23–27]. The cap4 methylations are vital for trans-splicing[23,28] and thus essential for the survival of the trypanosomal cell.

In opisthokonts, the eukaryotic clade grouping animals, fungi, and yeasts but not kinetoplastids, transcripts emerging from RNA polymerase II (RNAPII) harbor an $m^7$GpppN or $m^7$GpppNm modified cap (where N can be any nucleotide), termed cap0 or cap1, respectively. This $m^7$G cap is bound directly by the nuclear-cap-binding complex (CBC)[29–31]. Opisthokont CBC is a heterodimeric protein complex consisting of the small $m^7$G-binding subunit CBP20 (or NCBP2) and the larger scaffolding unit CBP80 (or NCBP1)[32–35]. The structural ensemble of both subunits is required for the cap-binding activity of the complex[32,36,37].

Opisthokont CBC has a crucial role in the maturation and fate of the RNAs. It mediates mRNA splicing[32] and 3′-end processing[38,39] and influences RNA localization[36,40]. The CBP80 subunit serves as a platform for the binding of various interaction partners that are required during biogenesis, cellular targeting, or nuclear export of different RNA species, e.g., NELF-E, ARS2, PHAX, NCBP3, FLASH, and ALYREF[41–50]. However, the targeting of defective RNA to destructive pathways also relies on CBC. For example, ZC3H18, ZFC3H1, or ZC3H4 connect CBP80 to the PAXT and NEXT complexes, which direct the RNP to the RNA degrading exosome[51–56]. Upon mRNA maturation, the mature CBC-bound mRNP is transported to the cytosol, and the CBC is replaced by the cytosolic cap-binding factor eIF4E[57,58]. Importin-α and importin-β bind the free CBC and shuttle it back to the nucleus where a new cycle can start[57,59,60].

In Trypanosoma brucei (T.brucei), the CBP20 subunit was identified through its high amino acid identity with the human protein (42%). In addition, the tyrosine residues that bind the cap are conserved[61]. Yet, instead of forming a dimer with a CBP80 homolog, T. brucei CBP20 (TbCBP20) was co-purified in a complex with four other proteins[61]. One of the identified binding partners was importin-α, but the three other factors, TbCBP30, TbCBP66, and TbCBP110, named by their molecular weights, lack homology to annotated proteins and are only found in the Trypanosomatidae family. RNAi knock-down of TbCBP20, TbCBP30, and TbCBP110 resulted in trans-splicing defects and was lethal for T. brucei cells[61]. Under these conditions, splicing precursors like the polycistronic RNA and SL RNA accumulated, and the levels of the Y-structured splicing intermediate were reduced, suggesting a critical role of TbCBC in an early step of trans-spliceosome assembly. Details about the interaction between TbCBC, the cap4 SL RNA, and the trans-spliceosome are currently lacking, and no function has been assigned to these three TbCBC subunits.

Here, we define the molecular architecture of TbCBC and its interactions with different capped and uncapped RNA species, based on cryo-EM structures and biochemical experiments. Our data reveal a bilobal structure of the trypanosomatid CBC. We confirm TbCBP20 as the cap-binding subunit and identify TbCBP110 as the homolog of opisthokont CBP80, despite their poor sequence conservation. TbCBP20 and TbCBP110 form the cap-binding core module, which can bind SL RNA independent of the cap4 modifications that are unique to kinetoplastids. TbCBP30 comprises mostly intrinsically unfolded regions and bridges the TbCBP20-TbCBP110 core to the TbCBP66 subunit, which contains an additional SL RNA interaction site with specificity for double-strand RNA. Our study will serve as a basis for further detailed dissection of RNA processing pathways in kinetoplastids, as CBC – in analogy to opisthokonts – may play a central role in mRNA biogenesis and localization.

## Results

### Expression and purification of different TbCBCs

To characterize the trypanosomatid CBC, we co-expressed and purified different combinations of T. brucei CBC subunits in a baculovirus expression system. We obtained pure and monodisperse samples of a tetramer (via the co-expression of TbCBP20-TbCBP110-TbCBP30-TbCBP66, 230 kDa), a trimer (via the co-expression of TbCBP20-TbCBP110-TbCBP30, 164 kDa) and a dimer (via the co-expression of TbCBP20-TbCBP110, 134 kDa) (Fig. 1B, C). Our inability to co-express or purify a trimer consisting of TbCBP20, TbCBP110, and TbCBP66 indicated that the expression of soluble TbCBP66 relied on the presence of TbCBP30. The purified dimeric, trimeric, and tetrameric TbCBC complexes were stable and monodisperse and thus suitable for biochemical experiments and cryo-EM analysis (Fig. 1C and Supplementary Fig. S1A, B).

### The cryo-EM structure of tetrameric TbCBC

To gain insight into the architecture of trypanosomatid CBC, we generated a cryo-EM structure of the TbCBC tetramer. Since an affinity assay indicated the ability of the tetramer and dimer to interact with the $m^7$GMP of the RNA 5′-cap (Fig. 1D and Supplementary Fig. S1C), we included the commercially available $m^7$GpppA in our cryo-EM grid preparations. The cryo-EM reconstruction extended to a resolution of 2.4 Å but accounted for only 48% of the residues in the complex. Although most of TbCBP20, TbCBP110, and an N-terminal portion of TbCBP30 could be modeled into the coulomb density, the C-terminal domain of TbCBP20, large parts of TbCBP30, and the entire TbCBP66 were absent (Fig. 1E, Supplementary Fig. S2, and Table 1). Nevertheless, mass photometry indicated the integrity of the sample at low concentrations (Supplementary Fig. S1B and Supplementary Table S1). Therefore, we assessed the disorder of the TbCBC tetramer in solution using small-angle X-ray scattering (SAXS). The radius of gyration of 6.15 ± 0.28 nm indicated the presence of the intact tetrameric complex, but the Kratky plot suggested disorder and/or flexible portions of the complex. In contrast, the TbCBP20-TbCBP110 dimer was globular and well-ordered with a radius of gyration of 3.27 ± 0.05 nm (Fig. 1F and Supplementary Fig. S3A). The reconstructed and modeled portions of the tetrameric complex comprise mainly the large, exclusively α-helical TbCBP110 subunit. At a concave surface of TbCBP110, the smaller TbCBP20 subunit is bound. The N-terminal domain and the central RNA recognition motif (RNP) of TbCBP20 are well-ordered, but the C-terminal domain of TbCBP20 is not resolved indicating its flexibility. The $m^7$GMP portion of the cap-analog is coordinated by TbCBP20 (Fig. 1G). An N-terminal peptide of TbCBP30 (residues R33-I82) is bound in the cleft formed between TbCBP20 and TbCBP110 and the protein continues by forming a large loop across the adjacent surface of TbCBP110. Coulomb density for TbCBP66 is absent. Overall, the cryo-EM data and SAXS measurements indicate that the CBC complex is bilobal, consisting of a core formed by TbCBP20-TbCBP110 and an N-terminal part of TbCBP30 to which the rest of TbCBP30 and TbCBP66 are flexibly tethered.

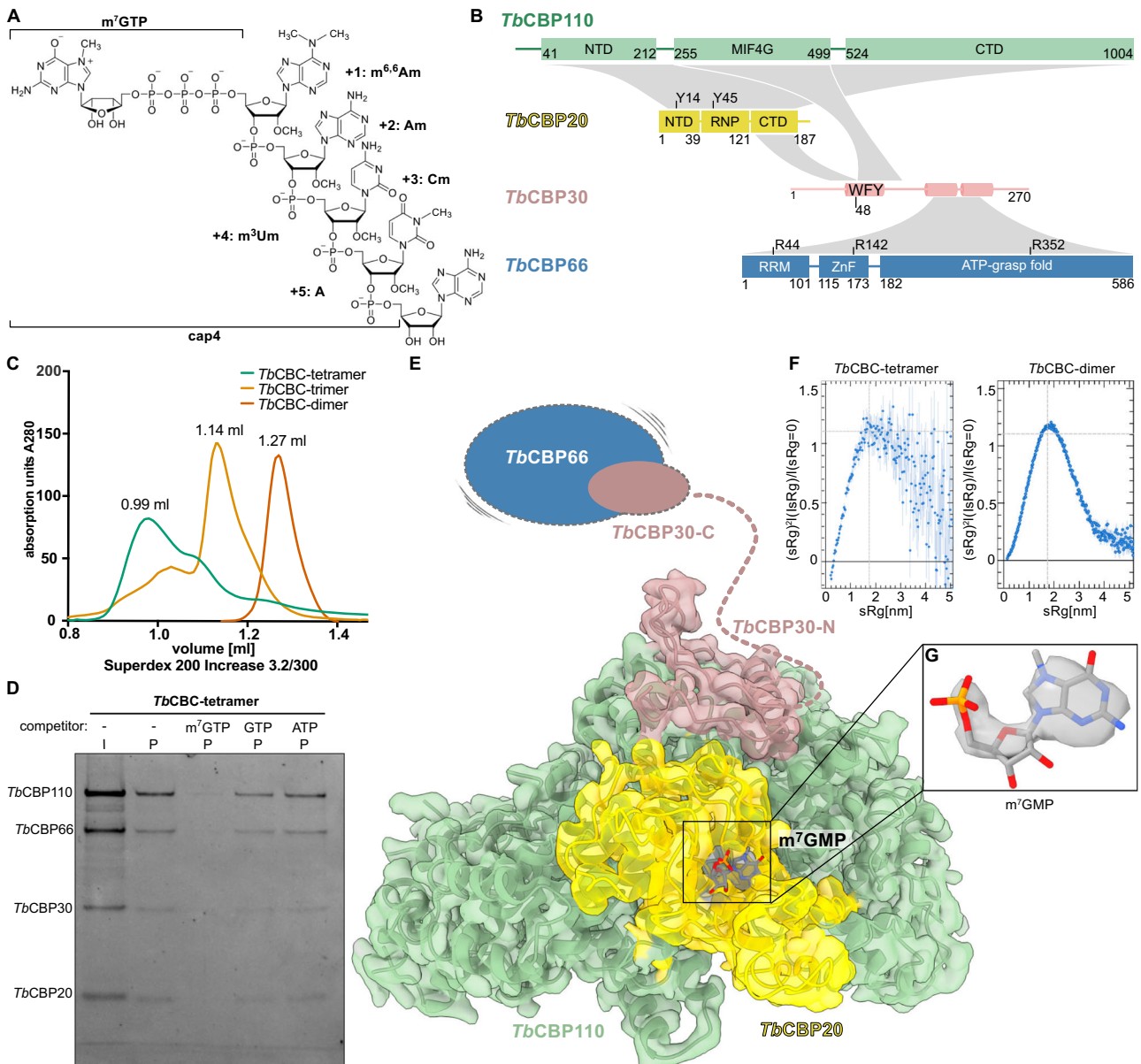

**Fig. 1 | Cryo-EM structures of the *T. brucei* cap-binding complex. A** Chemical structure of the 5'-end of the SL RNA with the cap4 modifications and m7G moiety indicated. **B** Domain organization of the *Tb*CBC subunits. *Tb*CBP110 (green), *Tb*CBP20 (yellow), *Tb*CBP30 (pink) and *Tb*CBP66 (blue). Structural information for *Tb*CBP110 and *Tb*CBP20 is based on cryo-EM data from this study. The domain organization of *Tb*CBP66 and the structural elements of *Tb*CBP30 are based on Alphafold2 structure predictions. Interactions are indicated in gray. Key functional residues are indicated. **C** Size exclusion chromatography profiles of different CBC complexes that could be obtained through co-expression: *Tb*CBC-tetramer (containing *Tb*CBP20, *Tb*CBP110, *Tb*CBP30, *Tb*CBP66), *Tb*CBC-trimer (containing *Tb*CBP20, *Tb*CBP110, *Tb*CBP30), *Tb*CBC-dimer (containing *Tb*CBP20, *Tb*CBP110). A Superdex 200 Increase 3.2/300 column was used, the data is representative for two or more runs. **D** Cap-binding assay on immobilized γ-Aminophenyl-m7GTP resin with the *Tb*CBC-tetramer and addition of different putative competitor molecules.

I = input, P = pull-down. A Coomassie-stained 12% SDS-PAGE is shown as a representative gel of three repetitions of the assay. **E** Cryo-EM reconstruction of the *Tb*CBC-tetramer at 2.4 Å. The DeepEMhancer post-processed map is shown at 0.03 level as a transparent surface containing the model in cartoon representation, m7GMP ligand as a stick model. Domains absent in the map but present in the sample (*Tb*CBP30-C and *Tb*CBP66) are indicated schematically. Colors as in (**B**). **F** Normalized Kratky plots obtained from Small Angle X-ray Scattering (SAXS) data of the *Tb*CBC-tetramer sample in comparison with the *Tb*CBC-dimer sample. The SEC-SAXS experiment was run once, the plot integrates the peak frames. Error bars indicate the standard deviation from the mean of the scattering pattern. **G** Cryo-EM map (DeepEMhancer post-processed) for m7GMP with surrounding residues as a stick model. The map is shown at 0.03 level. Source data are provided as a Source Data file.

### Trypanosome *Tb*CBP110 is the homolog of opisthokont CBP80

Due to the lack of sequence homology to characterized proteins, the function of *Tb*CBP110, the largest subunit of the assembly, was unclear. Our cryo-EM structure reveals a resemblance between the overall shape of *T. brucei* CBP110 and human CBP80, despite their low sequence identity of around 10 %. Mammalian CBP80 is composed of three consecutive α-helical domains connected through linkers; these domains are similar to MIF4G (middle domain of eIF4G) domains found in a variety of proteins involved in RNA metabolism. CBP80 and *Tb*CBP110 share this tripartite domain arrangement, and both have an exclusively α-helical secondary structure (Fig. 2A, B). The loops connecting the α-helices in *Tb*CBP110 are longer than in human CBP80,

**Table 1 | Cryo-EM data collection, refinement, and validation statistics**

| | *Tb*CBC-tetramer EMD-50173 PDB-9F3F | *Tb*CBC-trimer EMD-50217 PDB-9F67 |
|---|---|---|
| Data collection and processing | | |
| Magnification | 165 000 | 105 000 |
| Voltage (kV) | 300 | 300 |
| Electron exposure (e–/Å²) | 51.56 | 37.8 |
| Defocus range (µm) | −1 to −2.4 | −1 to −2.4 |
| Pixel size (Å) | 0.8127 | 0.42 |
| Symmetry imposed | C1 | C1 |
| Initial particle images (no.) | 6,907,011 | 3,042,614 |
| Final particle images (no.) | 251,312 | 69,764 |
| Map resolution (Å) | 2.4 | 2.8 |
| FSC threshold | 0.143 | 0.143 |
| Refinement | | |
| Initial model used (PDB code) | – | 9F3F |
| Map sharpening B factor (Å²) | − 42.6 | − 90.0 |
| Model composition | | |
| Non-hydrogen atoms | 7505 | 7948 |
| Protein residues | 939 | 981 |
| Ligands | 1 | 1 |
| B factors (Å²) | | |
| Protein | 48.67 | 61.58 |
| Ligand | 65.23 | 135.2 |
| R.m.s. deviations | | |
| Bond lengths (Å) | 0.008 | 0.009 |
| Bond angles (º) | 1.353 | 1.356 |
| Validation | | |
| MolProbity score | 2.31 | 1.93 |
| Clashscore | 16.31 | 13.35 |
| Rotamer outliers (%) | 2.45 | 0.12 |
| Ramachandran plot | | |
| Favored (%) | 95.67 | 95.77 |
| Allowed (%) | 4.33 | 4.23 |
| Disallowed (%) | 0 | 0 |

but are not resolved in our reconstructions; therefore, despite the higher molecular weight of *Tb*CBP110, the sizes of the models appear similar. The *Tb*CBP110 N-terminal domain does not resemble any annotated fold, judged by structural homology searches (Fig. 2C). The C-terminal domain (CTD) of *Tb*CBP110 resembles an MIF4G-like domain but contains two additional helical insertions (Fig. 2D). The middle domain of *Tb*CBP110 is an MIF4G-like domain with 15% sequence identity to the MIF4G-2 of CBP80 and their models superpose with an RMSD of 1.095 Å across 59 pruned atoms (6.133 across all 170 pairs) (Fig. 2E). A large concave groove spanning the middle MIF4G-like and CTD domains in *Tb*CBP110 accommodates the *Tb*CBP20 subunit comprising a surface of 2458.7 Å² (Pisa server v1.52). Similar to the human complex, mostly negatively charged residues in *Tb*CBP20 interact with complementary positive charges on *Tb*CBP110, resulting in electrostatic interactions and salt bridges. The similar overall features of the assemblies suggest that trypanosomatid CBP110 and CBP20 – like human CBP20-CBP80 – might form an entity and rely on each other for cap-binding (Supplementary Fig. S3B)[34,35,62]. Therefore, based on the structural features discussed above, we conclude that *Tb*CBP110 is the trypanosomatid homolog of mammalian CBP80.

## CBP20 is the conserved m⁷G interaction unit of kinetoplastid CBC

The sequence conservation of *Tb*CBP20 initially led to the identification of the *T. brucei* CBC[61]. Our data reveal the *Tb*CBP20 structure with a central RNP motif forming a β-sheet and additional N- and C-terminal domains, highly similar to its human homolog. The C-terminal domain of *Tb*CBP20 is unresolved in our maps. In human CBP20, parts of the N- and C-terminal domains are intrinsically unfolded and adopt a secondary structure only upon binding to the RNA cap[35,63]. A map reconstruction from discarded, incomplete particles of the *Tb*CBC-tetramer dataset, probably deficient of the m⁷G ligand, is highly anisotropic, but reveals a similar intrinsically unfolded behavior of the N- and C-termini of *Tb*CBP20, pointing to a conserved induced fit m⁷G binding mechanism (Supplementary Fig. S2).

### The structure of cap4-bound trimeric *Tb*CBC

To gain insight into the proposed cap4 interaction of *Tb*CBC, we prepared a sample with a synthesized cap4 hexa-nucleotide with the complete set of the kinetoplastid-specific modifications (m⁷Gppp$^{m6,6}$A$_m$pA$_m$pC$_m$pm³U$_m$pA) (Figs. 1A, 3A, and Supplementary Fig. S4). With the aim to improve the sample by eliminating flexible parts of the complex, we used only the trimeric *Tb*CBC assembly comprising *Tb*CBP20, *Tb*CBP110, and *Tb*CBP30. The reconstruction of the *Tb*CBC trimer extended to 2.8 Å and was overall very similar to the tetramer (RMSD of 0.442 Å across 719 pruned atom pairs) (Supplementary Figs. S5, S7A–C). Different from the tetramer, the trimeric sample resolved the C-terminal domain of *Tb*CBP20 and revealed further residues of the bound *Tb*CBP30 peptide. The three domains of *Tb*CBP20 are highly similar to human CBP20 (PDB:1H2T[35], superposing with an RMSD of 0.729 Å across 108 pruned atom pairs (Fig. 3A and Supplementary Fig. S7D).

In the cryo-EM maps of the CBC tetramer, we identify coulomb density accounting for the m⁷GMP moiety of the added cap analog (Fig. 1G). The trimeric sample resolves the m⁷GTP moiety of the cap4-hexa-nucleotide ligand in a similar arrangement in good resolution; the m⁷G binding resembles the arrangement in human CBP20 with conserved m⁷G coordination residues (Fig. 3A–C, and Supplementary Figs. S7D, S8). The nucleotide inserts into a pocket formed between the RNP and the NTD domains. Two conserved tyrosines, Y14 and Y45, sandwich the nucleobase via π-stacking[34,35,63]. The side chain carboxyl of aspartate D118 and the main chain carbonyl of tryptophan W117 form hydrogen bonds with the N2 amine group. Our nucleotide-competition pull-down assay (Fig. 1D) and nuclear magnetic resonance spectroscopy data of a saturated transfer difference (STD) experiment (Supplementary Fig. S6) indicate the specificity of *Tb*CBC for m⁷G compared to GTP and ATP, consistent with the preference of the *human* protein[62]. This is in line with data showing that m⁷GTP competed efficiently with cap4-SL RNA binding to *Tb*CBC, while $^{2,2,7}$GpppG and ApppG were inefficient competitors (Fig. 1D[61]. In addition, sugar and phosphate moieties of m⁷GTP are coordinated by the side chains of arginine R125, valine V136, arginine R129, and glutamine Q135, all conserved to the human protein. With a difference to human CBP20, the N1 and O6 atoms of m⁷G are not specifically coordinated; the coordination residues glutamate D114 and arginine R112 in human CBP20 are substituted through serine S116 and threonine T114, respectively (Fig. 3B, C). In conclusion, the comparison of human and *T. brucei* CBP20 reveals a highly similar m⁷G binding site, with a preference towards m⁷G and a similar mechanism of ligand-induced stabilization of the binding site.

The nucleotides following the m⁷GTP display significantly less resolution compared to the m⁷G moiety, indicating their partial flexibility; a weak signal for the nucleotides can only be obtained at a very reduced map level (Fig. 3E and Supplementary Fig. S7E). The C-terminal domain of *Tb*CBP20 lies in proximity to the nucleotide, but base- or modification-specific contacts, or positively charged residues

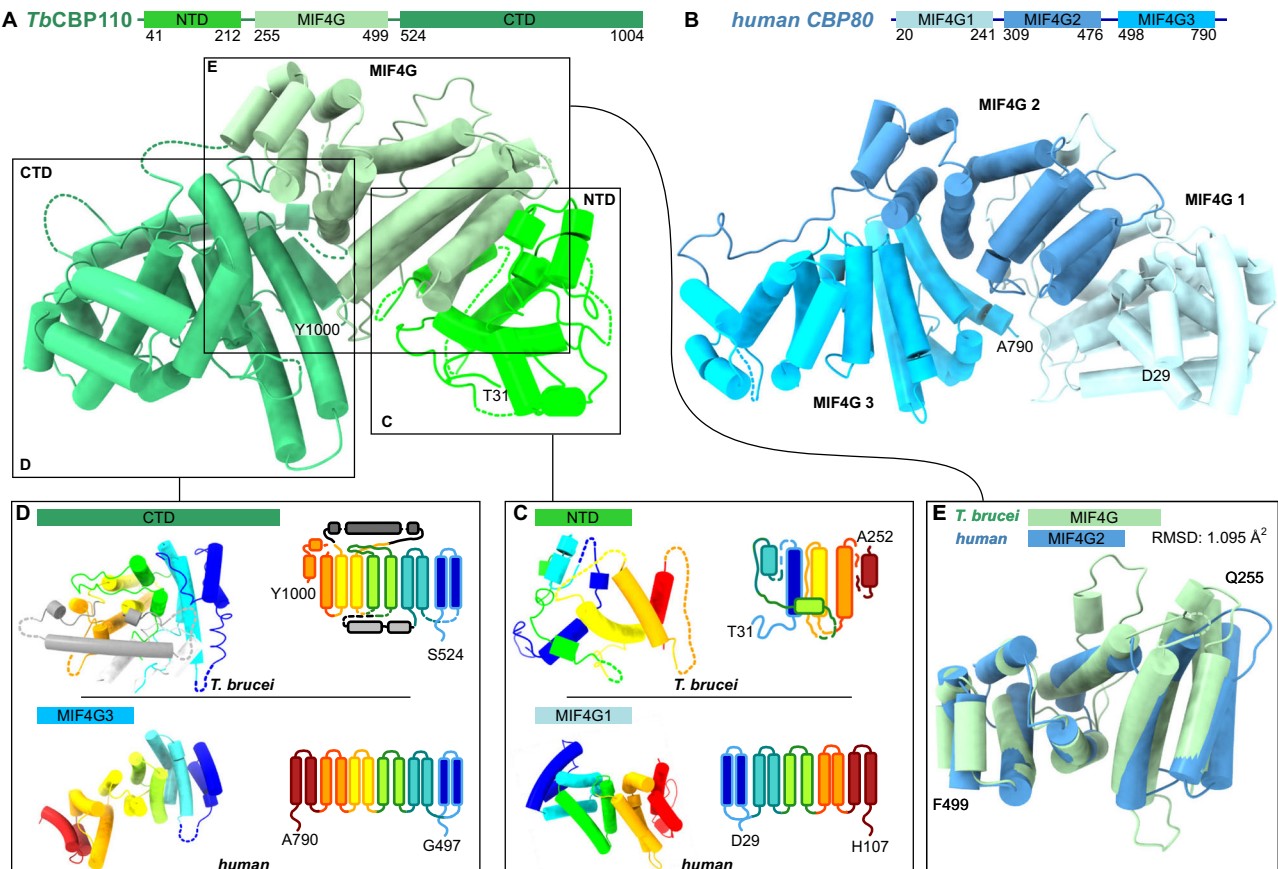

**Fig. 2 | The structure of *T. brucei* CBP110. A** *T. brucei* CBP110 is an α-helical protein with three consecutive domains: N-terminal (NTD), middle MIF4G-like, and C-terminal (CTD) domain. Cartoon representation with tubes representing α-helices. View similar to Fig. 1E, *Tb*CBP20, and *Tb*CBP30 models removed for clarity. *Tb*CBP20 would be positioned in front of the view. **B** Human CBP80 (PDB: 1h2t[35], with its three MIF4G-like domains has a similar shape and domain arrangement. **C** Comparison of *Tb*CBP110 NTD and human CBP80 MIF4G-1 in topological coloring (N-terminus of the domain in blue to C-terminus of the domain in red); cartoon model of the domains and topological schematic. **D** Comparison of *Tb*CBP110 CTD and human CBP80 MIF4G-3, as in (**C**). **E** Superposition of the *Tb*CBP110 MIF4G-like domain (green) and human CBP80 MIF4G2 (blue). The domains superpose with an RMSD of 1.095 Å across 59 pruned atom pairs.

that would coordinate the phosphodiester backbone of the RNA are missing. With the exception of glutamine Q142, *Tb*CBP20 exposes only hydrophobic residues (V136, V137, V140, L154) towards the nucleotide stretch. All these residues are conserved in trypanosomatids, implying that a tight cap4-coordination might have been evolutionarily disfavored. Overall, while our reconstructions confirm that *Tb*CBP20 is the conserved m$^7$GTP interaction subunit of the *T. brucei* cap-binding complex, they also indicate that it does not tightly coordinate cap4.

### *Tb*CBC binds the SL RNA independently of the kinetoplastid-specific cap4 modifications

To investigate the interaction between *Tb*CBC and SL RNA and its unusual cap4 structure in more detail, we conducted electrophoresis mobility shift assays (EMSAs). We generated capped oligonucleotides by a combination of chemical synthesis and enzymatical modification to compare *Tb*CBC binding to the 39 nt SL RNA mini-exon carrying different 5'-modifications: the uncapped 5'-OH-SL RNA exon (OH-SLe), the cap0 SL RNA exon (cap0-SLe) and the cap4 SL RNA exon (cap4-SLe) (Fig. 3F, G and Supplementary Fig. S9). The *Tb*CBC dimer and trimer do not strongly interact with OH-SLe, indicating that the presence of the m$^7$G-cap is a requirement for the interaction of the SL RNA with the *Tb*CBP20-*Tb*CBP110 core complex. However, both bind cap0 and cap4 modified SL RNA exon with similar affinity (Supplementary Table S2), indicating that the kinetoplastid-specific cap modifications are not necessary for the interaction. This is in line with our observations from the cryo-EM structures (Fig. 3A). In conclusion,

our data indicate that m$^7$G is sufficient for the interaction of *Tb*CBC with SL RNA and that neither the cap4 nucleotides nor their modifications are required.

### *Tb*CBP30 is the bridging subunit between the *Tb*CBP110-*Tb*CBP20 core and *Tb*CBP66

Next, we turned to the part of the CBC complex comprised of *Tb*CBP30 and *Tb*CBP66. Alphafold2 predicts *Tb*CBP30 to be largely unstructured with only a few secondary structure elements. This is consistent with the SAXS data indicating flexible parts of the complex and the absence of density for most of *Tb*CBP30 in our structures (Fig. 1F and Supplementary Fig. S10A)[64,65]. Nonetheless, coulomb density in our cryo-EM maps of the *Tb*CBC trimer and tetramer accounted for a portion of *Tb*CBP30 (Figs. 1E, 4A). A characteristic hydrophobic α-helix with large side chains in the N-terminal region of *Tb*CBP30 (residues P43 – G60) guided model building of *Tb*CBP30 into the cryo-EM maps. In this α-helix, four large hydrophobic residues (W48, F52, Y55, W59) orient towards *Tb*CBP110 and insert into a hydrophobic groove on the *Tb*CBP110 MIF4G-like domain. (Fig. 4A, B). We used transient overexpression of HA-tagged *Tb*CBC subunits in HEK293T cells followed by pull-down via mCherry-*Tb*CBP30 as bait to assess complex formation with mutant protein. The conserved hydrophobic residues are important for complex formation since single-point mutations of W48, F52, and Y55 interfere with the association of *Tb*CBP30 to the *Tb*CBP20-*Tb*CBP110 complex in the co-precipitation assay with overexpressed

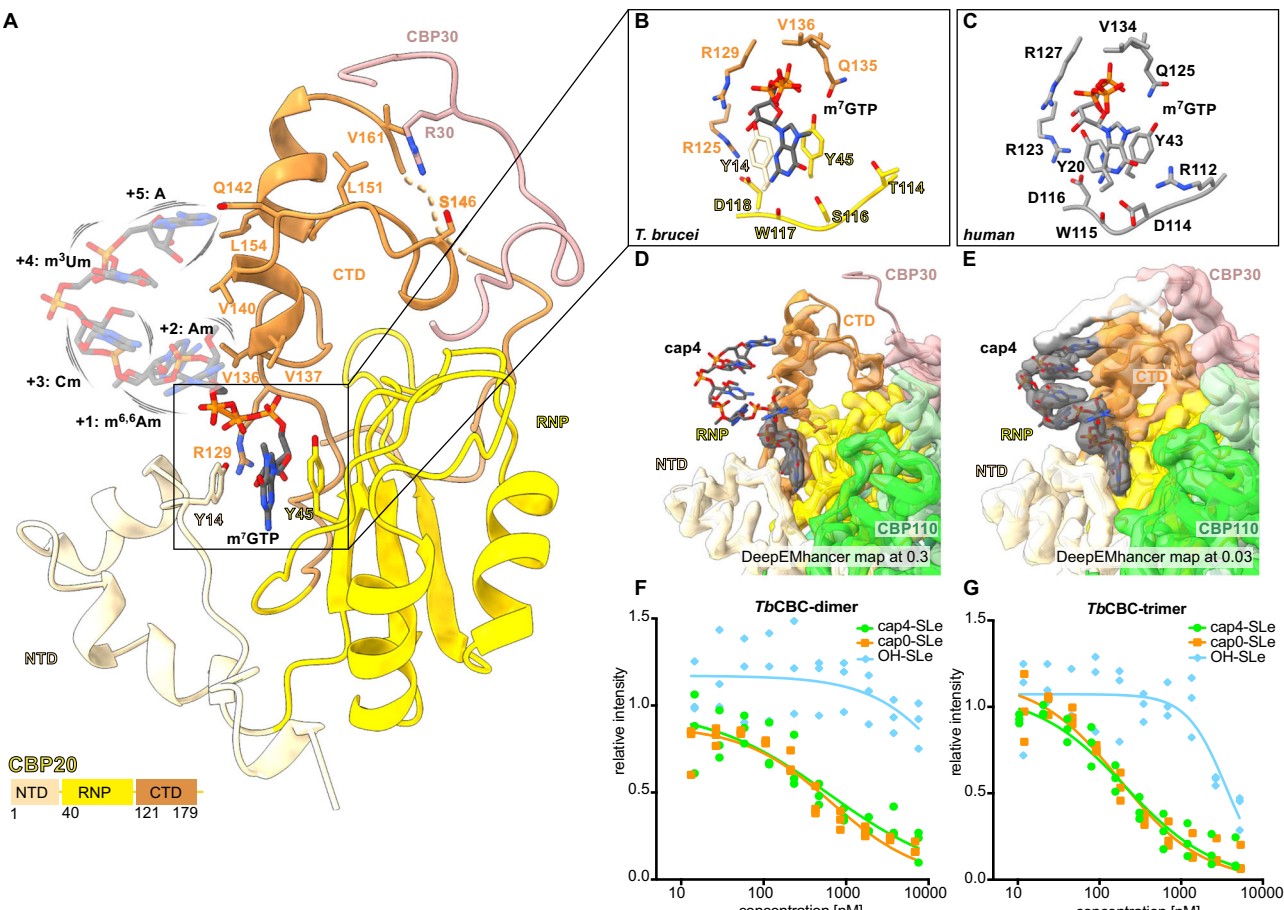

**Fig. 3 | RNA cap coordination in *T. brucei* CBP20. A** Cartoon representation of *Tb*CBP20 of the *Tb*CBC-trimer-cap4 sample in shades of yellow indicating the three domains of *Tb*CBP20. The *Tb*CBP30 peptide bound to the CTD of *Tb*CBP20 is also shown, with contact residues in stick representation. The cap-ligand is shown as a stick model, and its flexibility is indicated. **B** Residues in the m7GTP binding pocket of *T. brucei* and (**C**) human CBP20, based on PDB:1H2T[35]. **D** Deep-EMhancer post-processed trimer cryo-EM map transparent with cartoon model and cap4-hexanucleotide in stick representation and at level 0.3 and (**E**) at a very reduced level of 0.03. **F** Electrophoresis mobility shift assay (EMSA) of *Tb*CBC-dimer binding to the SL RNA exon with different 5′ chemistry. **G** EMSA of *Tb*CBC-trimer binding to the SL RNA exon with different chemistry. EMSAs in (**F**) and (**G**) were run in triplicate reactions; the intensity of PAGE gel bands was normalized in relation to the protein-free RNA control band. A Hill model was used to fit the binding curve. Source data are provided as a Source Data file.

*Tb*CBC subunits. The mutation of W48 had the strongest effect, while W59 seemed less important for the interaction (Fig. 4C and Supplementary Fig. S10B). A similar interaction via a tryptophan-containing hydrophobic α-helix can be observed in a set of transient mammalian CBC co-factors, including NELF-E, NCBP3, PHAX, and ZC3H18 (Supplementary Fig. S10C)[43].

In addition to this hydrophobic α-helix, the N-terminal portion of *Tb*CBP30 (S27 – P42) binds to the *Tb*CBP20 C-terminal domain via polar interactions of two arginines of *Tb*CBP30 (R30 and R33, R30 is conserved) and then continues in a cleft formed between *Tb*CBP20 and *Tb*CBP110 (Figs. 3A, 4D). In this region, arginine R35 of *Tb*CBP30 binds into a deep pocket formed between *Tb*CBP20 and *Tb*CBP110 and is stacked between two tyrosines, one from each subunit (Y52 of *Tb*CBP20 and Y479 of *Tb*CBP110). This mode of arginine coordination strongly resembles the interaction of the human CBC with the co-factors NELF-E and ARS2; in fact, this particular arginine has been determined to be crucial for their interaction with the CBC (Fig. 4D and Supplementary Fig. S10D)[41,42]. However, the mutation of R35 in *Tb*CBP30 to serine/lysine had no effect on its association with the CBC (Fig. 4C and Supplementary Fig. S10B). The affinity of a TAMRA-labeled *Tb*CBP30 peptide comprising residues R34-G60 towards the *Tb*CBP20-*Tb*CBP110 dimer was determined to be 980 nM ± 386 nM, by microscale thermophoresis (Fig. 4E and Supplementary Fig. S11). C-terminal to the hydrophobic helix, the *Tb*CBP30 peptide bends back towards

*Tb*CBP20 in a large loop across the *Tb*CBP110 middle domain, but the slightly lower resolution in this region indicates that this interaction might be weaker compared to the rest of *Tb*CBP30 (Fig. 4A and Supplementary Fig. S5). Overall, through our cryo-EM structure and interaction experiments, we identify a hydrophobic α-helix in the N-terminus of *Tb*CBP30 with a critical tryptophan residue as a crucial interaction site within the kinetoplastid CBC complex. Here, distinct interaction features resemble the binding of mammalian CBC co-factors.

The *Tb*CBP30 hydrophobic α-helix single-point mutants that disrupt the interaction of *Tb*CBP30 with the *Tb*CBP20-*Tb*CBP110 dimer still bind *Tb*CBP66, adding evidence that parts of *Tb*CBP30 and *Tb*CBP66 form an independent unit (Fig. 4C). This is in line with our observation that the solubility of *Tb*CBP66 in the co-expression experiments relied on the presence of *Tb*CBP30. To define the section of *Tb*CBP30 that interacts with *Tb*CBP66, we conducted further co-precipitation experiments with N- and C-terminal truncations of affinity-tagged *Tb*CBP30 overexpressed together with *Tb*CBP66. Our data reveal that a region within the C-terminus of *Tb*CBP30 (residues K150 to A200) is required for the co-purification of full-length *Tb*CBP66 (Fig. 4F). Together, these data indicate that *Tb*CBP30 acts as a flexible linker that interacts via an N-terminal region (residues P43 – G60) with the *Tb*CBP20-*Tb*CBP110 core, and via a C-terminal region (within K150-A200) with the *Tb*CBP66 subunit.

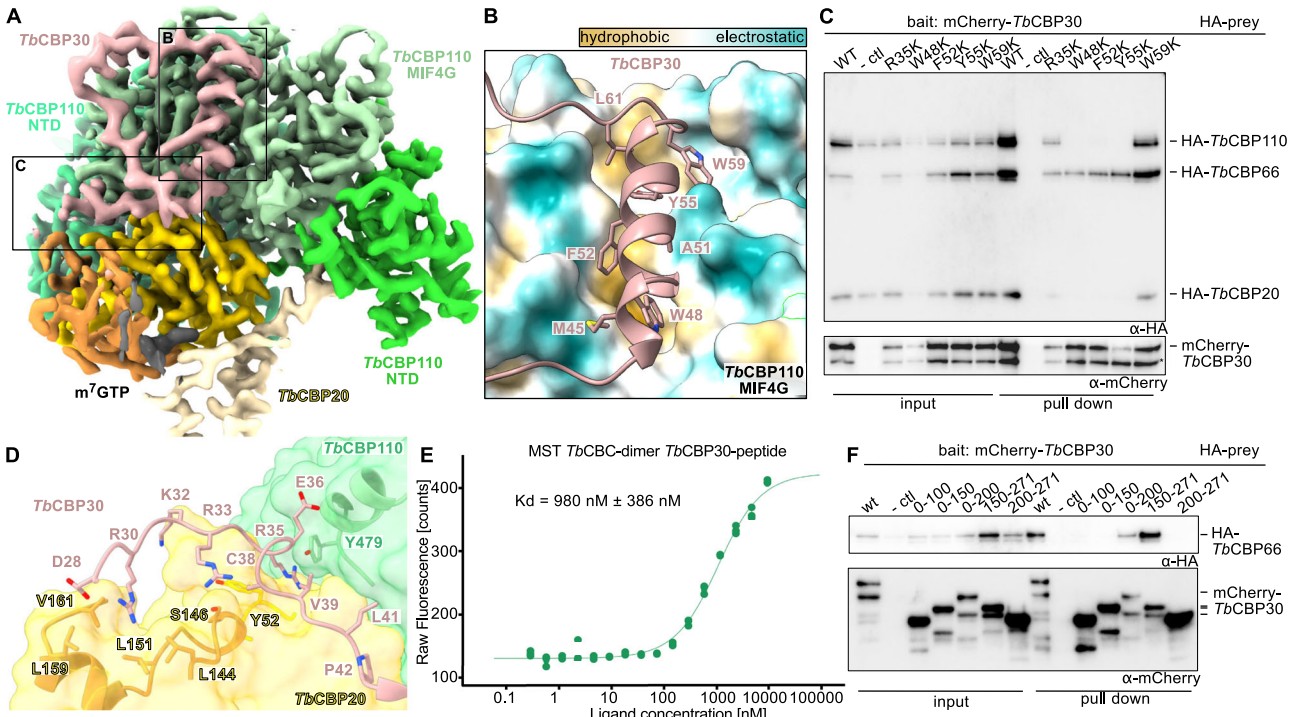

**Fig. 4 | *Tb*CBP30 is the bridging subunit between the *Tb*CBP20-*Tb*CBP110 core and *Tb*CBP66. A** DeepEMhancer post-processed map of the *Tb*CBC-trimer-cap4 sample is shown at level 0.3. Colors as in previous figures. **B** A hydrophobic helix of *Tb*CBP30 (in cartoon representation) binds to a hydrophobic groove on *Tb*CBP110 (surface representation with ChimeraX hydrophobicity coloring)[90]. **C** Co-precipitation assay of co-expressed HA-tagged *Tb*CBP20, *Tb*CBP110, *Tb*CBP66 with mCherry-*Tb*CBP30 mutants and Western blot detection. A representative Western blot of more than three experiments is shown. **D** Binding of N-terminal portions of the *Tb*CBP30 peptide to *Tb*CBP20 and into the groove between *Tb*CBP20 and *Tb*CBP110. *Tb*CBP20 and *Tb*CBP110 are transparent surface representations with *Tb*CBP30-facing residues as cartoons and sticks. *Tb*CBP30 in cartoon representation, with interaction residues as sticks. **E** Fluorescence data of a microscale thermophoresis experiment determining the interaction between the *Tb*CBC-dimer and TAMRA-labeled CBP30(R34-G60) (*n* = 3). **F** Co-precipitation assay of *Tb*CBP66 with mCherry-CBP30 truncations western blot detection. A representative western blot of more than three experiments is shown. Source data are provided as a Source Data file.

## *Tb*CBP66 binds the SL RNA through RNA double-strand features

To determine a potential functional contribution of the *Tb*CBP66 subunit to the CBC complex, we tested the binding of the *Tb*CBC tetramer to capped and uncapped SL RNA using EMSAs. In contrast to the dimer and trimer (Fig. 3F, G), the *Tb*CBC tetramer binds SL RNA independent of the m⁷G modification, suggesting an additional RNA-binding site in the *Tb*CBP66 subunit (Fig. 5A). To express soluble *Tb*CBP66, we fused two *Tb*CBP30 Alphafold2-predicted α-helices (G141-N220) of the identified interaction region to the N-terminus of full-length *Tb*CBP66. This engineered *Tb*CBP66* construct yielded a soluble and monodisperse sample (Fig. 5B and Supplementary Fig. S1A, B). We then assessed the affinity of *Tb*CBP66* to the 5′-OH SL RNA exon used earlier (39 nt) (Figs. 3F, G, 5A), a 15 nt single-stranded RNA originating from a stretch in the 5′-end of the SL RNA, and the isolated stem-loop region of the SL RNA exon (Fig. 5C). The full SL RNA exon had a similar affinity to *Tb*CBP66* as the SL RNA exon stem-loop, but single-stranded RNA did not bind. This indicates that *Tb*CBP66 might specifically interact with double-stranded regions of the SL RNA. To cross-validate the identified RNA-binding activity of *Tb*CBP66, we used an Alphafold2 model of *Tb*CBP66 to identify positive surface patches that might play a role in RNA binding. The predicted model indicated three independent domains: an N-terminal RNA recognition motif (RRM), a Zinc finger domain, and an ATP grasp domain[64–66] (Fig. 5D). We mutated single positive surface residues of each domain to the inverse charge and tested RNA binding to the full SL RNA exon (Fig. 5E, Supplementary Fig. S12 and Supplementary Table S3). Single point mutants in the RRM domain (R44E) and Zinc finger (R142E) impaired RNA binding, confirming the role of the *Tb*CBP66 in the SL RNA interaction. The mutation in the ATP grasp domain (R352E) did not

have a strong effect. These data show that *Tb*CBP66 acts as an additional RNA binding subunit in the trypanosomatid CBC that interacts with double-strand features of the SL RNA.

## Discussion

In this study, we present the architecture of the tetrameric trypanosomatid nuclear cap-binding complex and determine how it interacts with the SL RNA. In analogy to opisthokont CBP20-CBP80, *Tb*CBP20 and *Tb*CBP110 form the cap-binding core of the complex with specificity for the m⁷GTP RNA cap. The *Tb*CBP66 subunit is flexibly tethered to this core by *Tb*CBP30 and provides a second RNA binding site with specificity for double-stranded RNA features of the SL RNA (Fig. 6). *Tb*CBC may bind the SL RNA early after capping, mediate trans-splicing and escort the pre-mRNA throughout its maturation in the nucleus, until the export of the mature mRNP to the cytosol. Therefore, *Tb*CBC represents a key component of kinetoplastid RNA biogenesis pathways. Understanding its structure and function will be the basis for the detailed dissection of kinetoplastid RNA metabolism.

An earlier study suggested that the kinetoplastid CBC has a high affinity for cap4-modified SL RNA[61]. While our data do not confirm this conclusion, our discovery of a second SL RNA binding site in *Tb*CBP66 shines a new light on the results previously reported. In the prior study, full-length 140 nt cap4 SL RNA, purified from *T. brucei* cells, was pre-incubated with *Tb*CBC and either the same full-length cap4 SL RNA or the di-nucleotide cap-analog m⁷GpppG was used as a competitor substrate. Based on our results, the double-stranded SL RNA portions in the longer substrate bind to *Tb*CBP66 and augment the affinity of *Tb*CBC for the full-length cap4 SL RNA rather than the cap configuration. In line with the prior study, we determined the affinity of

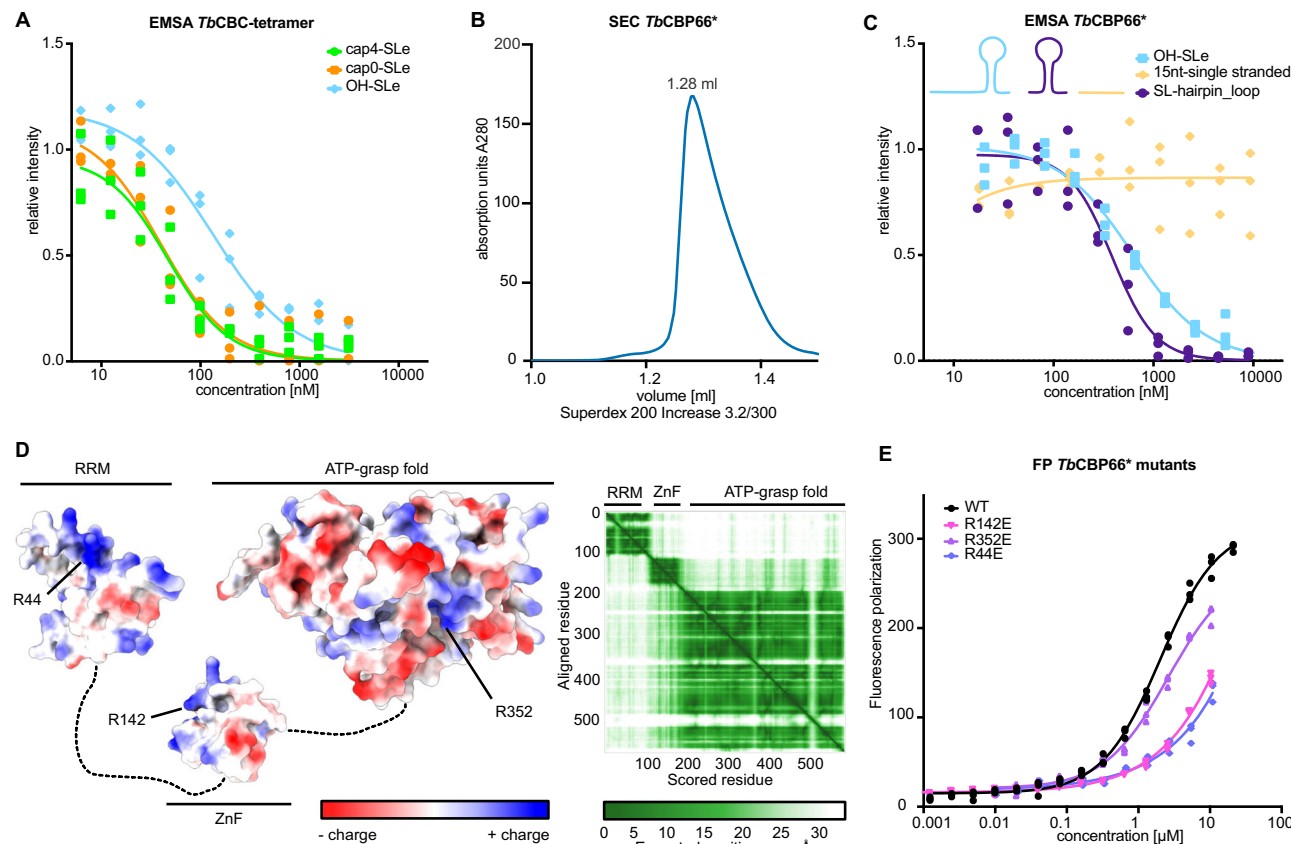

**Fig. 5 | *T. brucei* CBP66 is the dsRNA-binding subunit of *Tb*CBC. A** EMSA of *Tb*CBC-tetramer binding to the SL RNA exons with different 5' chemistry. **B** Size exclusion chromatography of the engineered *Tb*CBP66* (CBP30(G141-N220)-CBP66 fusion), a representative curve of more than 3 experiments is shown. **C** EMSA probing the interaction of *Tb*CBP66* with different uncapped RNAs: the 39 nt SL RNA exon (same ligand as in Fig. 3F, G), a 15 nt single-stranded RNA (A1-U15 of the SL RNA), and the 25 nt SL RNA hairpin loop comprising A14-G39 of the SL RNA. EMSAs in (**A**) and (**C**) were run in triplicate reactions, and the intensity of PAGE gel bands was normalized in relation to the protein-free RNA control band; A Hill model was used to fit the binding curve. **D** The Alphafold2 model of *Tb*CBP66 indicates three domains with exposed positive charges on all domains. A surface representation of the Alphafold2 model and the Predicted aligned error (PAE) plot indicating the domain position confidence are shown. **E** Fluorescence polarization (FP) assay probing the binding of the 39 nt SL RNA exon to *Tb*CBP66* single point mutants. The assay was run in triplicate and fit with a nonlinear sigmoidal fit model. See also Supplementary Fig. S12. Source data are provided as a Source Data file.

tetrameric *Tb*CBC towards cap4-SL RNA to be 47 nM, which lies in an acceptable range, given the different methodologies used, with this prior data determining the affinity to 26 nM[61].

The cap4 modifications are crucial for kinetoplastid RNA processing and trans-splicing[23,28,67], but our study suggests that *Tb*CBC is not the primary interaction partner of these modifications. Nevertheless, the knock-down of CBC components leads to defects in transsplicing[61]. This suggests that both cap4 and the CBC are involved in this process, but further investigations will be needed to reveal the molecular basis for how CBC modulates trans-splicing.

The cryo-EM structures of the *Tb*CBC trimer and tetramer allowed us to identify the binding site of *Tb*CBP30 to *Tb*CBP20-*Tb*CBP110. Interestingly, the binding motifs resemble the binding mode of transient mammalian CBC interactors, which bind in a modular, sometimes mutually exclusive, and sequential manner to influence the fate of the nascent RNP. A set of these co-factors interacts with CBC via a hydrophobic helix that contains a crucial tryptophan residue, which binds to a hydrophobic groove on the surface of CBP80. The mammalian co-factors have a central role in RNA metabolism and include NELF-E, a factor that binds the nascent mRNP and causes RNAPII transcriptional pausing; NCBP3, which facilitates splicing and nuclear export of the mRNP; PHAX, which is required for snRNA export; and ZC3H18, which directs the RNA for degradation[43] (Supplementary Fig. S10D). Interestingly, the early association of NELF to the CTD of RNAPII enables RNA capping. The interaction of the NELF subunit

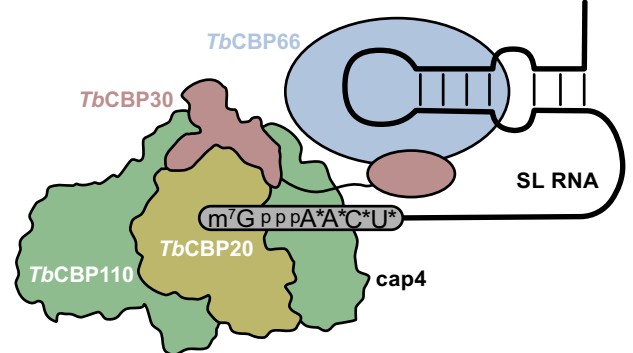

**Fig. 6 | Model of the tetrameric kinetoplastid cap-binding complex with SL RNA.** Colors as in Fig. 1B.

NELF-E with CBP80 most likely recruits the CBC to the freshly capped transcript. At what point and via which mechanism *Tb*CBC is recruited to the capped SL RNA is so far unresolved, but the mechanism may differ from opisthokonts due to kinetoplastid-unique polycistronic transcription and SL RNA trans-splicing. For example, RNAPII CTD phosphorylation, which is required for capping in opisthokonts, seems dispensable for co-transcriptional m⁷G capping in trypanosomatids[68,69]. Interestingly, all binding grooves on the surface

of mammalian CBP20-80 that bind NELF-E are also occupied in our *Tb*CBP20-*Tb*CBP110 models: the proximal binding groove and the tryptophan-containing helix are bound by the *Tb*CBP30 peptide, and the position defined as distal binding groove in mammalian CBC is covered by a peptide of *Tb*CBP20 itself (Supplementary Fig. S10C). Affinities of the mammalian interaction peptides are in the 110-200 nM range (173 nM for NCBP3, 125 nM for PHAX), while we determined the affinity of the *Tb*CBP30 portion covering the hydrophobic and proximal groove to be 980 nM. The measured affinity raises the question of whether the *Tb*CBP30-*Tb*CBP66 entity – in homology to the mammalian interactors – would only transiently interact with the *Tb*CBP20-*Tb*CBP110 core; however, the C-terminal portion of *Tb*CBP30, which was omitted in our experiment, may convey additional affinity. A very stable complex is indicated by our mass photometry data measured at low concentrations; further, *Tb*CBC resisted a high salt concentration washing step (1 M KCl) during the purification. These observations indicate high stability of the complex and would support that tetrameric CBC forms a permanent assembly. Further research is required to address this question.

Human CBC is a validated drug target, targeted by the drug obefazimod, which is currently in stage three clinical trials for the treatment of inflammatory bowel disease[70]. This indicates that CBC could serve as a therapeutic target also in Trypanosomatids, but the species-specificity of a potential drug must be assured. While the cap-binding site is not sufficiently divergent between the parasite and mammalian host CBC, other surfaces, like the RNA-binding site in *Tb*CBP66 may be sufficiently divergent to be used for species-specific anti-parasitic drugs. The RNA processing machinery of *T. brucei* is the target of the drug acoziborole (SCYX-7158), which is effective against human African trypanosomiasis (HAT), demonstrating that drugging the RNA metabolism of trypanosomes is an effective strategy[71,72]. This data fosters hope that molecules inhibiting the function of the trypanosomatid CBC could serve as anti-infectives for treating trypanosomatid diseases in the future.

## Methods

### Protein expression and purification

Different combinations of full-length *Tb*CBC subunits (Supplementary Table S4)[73,74] were cloned into pLIB and recombined into pBIG1a for expression in Hi5 cells using the Bigbac insect cell expression system[75,76]. Viruses for these constructs were generated according to ref. 77. For protein expression, Hi5 cells were infected with 0.5 % of V1 and incubated for 72 h at 27 °C. 2 L cell culture was pelleted at $2000 \times g$ and stored at − 80 °C for further use. For purification, a cell pellet was thawed on ice and resuspended in 50 ml of ice-cold lysis buffer (20 mM HEPES pH 7.5, 150 mM NaCl, 5 mM MgCl₂, 20 mM Imidazole, 2% Glycerol, 1 mM phenylmethylsulfonyl fluoride (PMSF), 2 μg/ml DNase), and lysed on ice by sonication (10 min, 30 % amplitude, 5 s on, 10 s off; Vibracells, Sonics). The total cell lysate was clarified by centrifugation at $40,000 \times g$ for at least 45 min at 4 °C. The clarified lysate was filtered (5 μm pore size) and loaded onto a pre-equilibrated 5 ml HisTrap HP affinity chromatography column (Cytiva) using a peristaltic pump. The column was washed with 10 column volumes (CV) of high salt buffer (20 mM HEPES, 1 M KCl, 5 mM MgCl₂, 20 mM Imidazole) followed by a wash with 10 CV of low salt buffer (20 mM HEPES, 50 mM NaCl, 5 mM MgCl₂, 50 mM Imidazole). The complexes were eluted with 50 ml of elution buffer (20 mM HEPES, 50 mM NaCl, 5 mM MgCl₂, 500 mM Imidazole), and loaded in a pre-equilibrated 5 ml HiTrap Heparin HP affinity column (Cytiva). The heparin column was first washed with a low salt buffer (20 mM HEPES, 50 mM NaCl, 5 mM MgCl₂) to remove imidazole, followed by elution via an increasing salt gradient to 1 M NaCl over 20 CV. The heparin elution was dialyzed overnight into a lower salt buffer (20 mM HEPES, 50 mM NaCl, 5 mM MgCl₂), concentrated to 2 mg/ml using an Amicon Ultra centrifugal filter (Milipore) with a corresponding MW cut-off, aliquoted into 60 μl aliquots with 5% glycerol, flash frozen in liquid nitrogen and stored at − 80 °C for further use.

### Mass photometry

A Refeyn OneMP mass photometer was used to determine the molecular weight of *Tb*CBC subcomplexes. Experiments were performed in SEC-Buffer (20 mM HEPES pH 7.5, 50 mM NaCl, 5 mM MgCl₂, 0.5 mM tris(2-caroxyethyl)phosphine (TCEP). A native protein ladder was used to calibrate the machine and conduct contrast-to-mass conversion to determine MW. 2 μL of protein (50−200 nM final concentration) was added to 18 μL of SEC-Buffer. Protein concentrations were adjusted to 1000−3000 counts. Movies of 120 s were recorded and analyzed using DiscoverMP (Refeyn).

### Cryo-EM sample preparation

A 2-fold excess of either m⁷G(5')ppp(5')A RNA Cap Structure Analog (New England Biolabs) or chemically synthesized cap4 hexa-nucleotide (see cap4 synthesis below) was added to the protein complex and incubated for 30 min. The sample was further purified by size exclusion chromatography (SEC) (20 mM HEPES 7.5, 50 mM NaCl, 5 mM MgCl₂, 0.5 mM TCEP) using a Superdex™ 200 Increase 3.2/300 column (Cytiva). A PELCO easiGlow (Ted Pella) was used to glow-discharge Quantifoil Gold 1.2/1.3 grids for 45 s at 30 mA on both sides. The grids were vitrified by plunge freezing in liquid ethane using a Vitrobot MARK IV (FEI) under the following conditions (Blot time: 2 sec, Blot Force: − 8, Volume: 1.5 μl applied on both sides, Temperature: 4 °C, Humidity: 100 %).

### Cryo-EM data acquisition

Micrographs for *Tb*CBC-tetramer were collected on a Titan Krios FEI (Thermo Fisher Scientific) at EMBL Heidelberg, equipped with a Gatan K2 camera operated at 300 kV. A magnification of 165,000x was used, resulting in a pixel size of 0.8127 Å/pixel. Automated data acquisition was carried out in Serial EM[78], with a total dose of 51.56 e⁻/Å² and 40 frames per movie, with a defocus range of − 1.0 μm to − 2.5 μm.

Micrographs for *Tb*CBC-trimer were collected on a Titan Krios FEI (Thermo Fisher Scientific) at ESRF in Grenoble, equipped with a Gatan K3 camera and operated at 300 kV in super-resolution mode. Micrographs were acquired at a magnification of 105,000x, resulting in a pixel size of 0.42 Å/pixel. Automated data acquisition was carried out in EPU (Thermo Fisher Scientific), with a total dose per movie of 37.8 e⁻/Å² and 40 frames per movie, with a defocus range of − 1.0 μm to − 2.4 μm.

### Cryo-EM data processing

For the *Tb*CBC-tetramer dataset, movies were motion corrected in Relion 3.1[79], followed by CTF estimation using CTFFIND-4.1[79–81]. Corrected micrographs were imported into WARP[82] for particle picking, in which 6 907 011 particles were picked using WARP's BoxNet2-Mask_21080918 model. Particles were binned twofold, extracted in RELION 3.1, and imported into CryoSPARC[83] for particle sorting. Particle sorting was done exclusively using 3D classification by generating 7 classes via ab initio reconstruction followed by heterogeneous refinement, after which all obtained classes that contained density for the N-terminal region of CBP20 were pooled back for further classification. Another cycle of ab initio reconstruction and heterogeneous refinement was carried out with 5 classes, followed by three rounds using 3 classes and one round with 2 classes, which reduced the number of particles to 251 312. These particles were imported in RELION for 3D refinement, followed by particle re-extraction, for which we Fourier-cropped the particles to the physical pixel size of 0.8127 Å/pixel. Three rounds of CTF-refinement followed by Bayesian polishing[84] and 3D refinement were carried out until a final overall resolution of 2.4 Å was reached.

For the *Tb*CBC-trimer dataset, movies were motion-corrected in RELION 3.1[79], followed by CTF estimation using CTFFIND-4.1[79–81]. Corrected micrographs were imported into WARP[82] for particle picking, in which 3,042,614 particles were picked using WARP's

BoxNet2Mask_21080918 model. Particles were binned twofold, extracted in relion 3.1, and imported into CryoSPARC[83] for particle sorting. Particle sorting was done exclusively using 3D classification by generating 7 classes via ab initio reconstruction followed by heterogeneous refinement, after which all obtained classes that contained density for the N-terminal and C-terminal region of CBP20 were pooled back for further classification. Another cycle of ab initio reconstruction and heterogeneous refinement was carried out with 7 classes, followed by a round with 5 classes. A final round was carried out with 3 classes, from which two where pooled back together and which reduced the number of particles to 369 417. These particles were imported in Relion for 3D refinement, followed by particle re-extraction, for which we Fourier-cropped the particles to the physical pixel size of 0.84 Å/pixel. Three rounds of CTF refinement followed by Bayesian polishing and 3D refinement were carried out until a final overall resolution of 2.5 Å was reached. Particles were further sorted by 3D variability analysis in CryoSPARC. Particles with the strongest density for the C-terminal domain of *Tb*CBP20 and cap4 RNA were pooled, and a final 3D reconstruction of 69 764 particles resulted in an overall resolution of 2.8 Å.

## Model building

*Tb*CBC-tetramer bound to the Cap Structure Analog model was built de novo in Coot 0.9.3[85,86]. The map was blurred to a value of 200 using the Coot:Sharpen/Blur Map tool of the cryo-EM module. Polyalanine helices were placed manually on the map to get the first model. Phenix.map_to_model[87] was used to fit the sequence register automatically, and the remaining sequence was manually fitted. The resulting model was validated through phenix.real_space_refinement[88]. The final model contains residues 1–128 and 171–180 from *Tb*CBP20, residues 31–63, 87–127, 139–167, 177–193, 212–225, 240–350, 362–449, 454–449, 454–499, 524–577, 599–665, 679–714, 732–792, 829–859, 865–892, 898–968 and 977–1000 form *Tb*CBP110, and residues 33–82 of *Tb*CBP30.

*Tb*CBC-trimer model was built based on the *Tb*CBC-tetramer model. The final model contains residues 1–162 and 167–179 from *Tb*CBP20, residues 41–63, 87–127, 139–167, 178–193, 212–252, 240–350, 362–499, 524–577, 599–665, 679–714, 732–792, 829–859, 865–893, 898–968 and 977–1000 form *Tb*CBP110, and residues 25–83 of *Tb*CBP30. The cap4 analog was built based on the structure of *Tc*EIF4E5 in complex with cap4 (PDB: 6O7Y[89],). ChimeraX was used for visualization and figure production[90].

## Cap-binding assay

The sample buffer was exchanged to cap-binding buffer (20 mM HEPES pH 7.5, 200 mM NaCl, 5 mM MgCl₂, 5 mM DTT, 0.05 % NP-40) using a Zeba™ Spin Desalting Column with a 7k MWCO (Thermo Fisher Scientific). For each reaction, 7 μg of protein was diluted in 500 μL of cap-binding buffer (20 mM HEPES pH 7.5, 200 mM NaCl, 5 mM MgCl2, 5 mM DTT, 0.05 % NP-40) supplemented with 0.7 μmol of competitor nucleotide (m⁷GTP, GTP or ATP, Jena Bioscience). Samples were incubated with 10 μL of equilibrated immobilized γ-Aminophenyl-m⁷GTP (C10-spacer) resin (Jena Bioscience) for 1 h at 4 °C. After incubation, the beads were washed three times with 500 μL of cap-binding buffer supplemented with 70 μmol of competitor nucleotide, resuspended in SDS-loading dye, incubated for 10 min at 95 °C and analyzed on SDS-PAGE.

## Small angle x-ray scattering (SAXS)

SEC-SAXS experiments were performed on beamline ID30A-1 at ESRF Grenoble. The system was coupled to Superdex™ 200 Increase 3.5/300 column (Cytiva), equilibrated in SEC-Buffer (20 mM HEPES / 50 mM NaCl / 5 mM MgCl₂ / 0.5 mM TECEP). For each measurement, 50 μL of the sample at 2 mg/ml were incubated with cap analog m⁷GpppA and injected into the column. Runs were carried out at room temperature. The buffer subtractions and further analysis were carried out in chromixs from the Atsas software suite[91].

## Co-precipitation experiments

For co-precipitation experiments, HEK293T cells were grown in DMEM high glucose media (Thermo Fisher Scientific), completed with 10 % FBS (Capricorn Scientific), 0.1 μg/ml Streptomycin and 0.1 Units/ml Penicillin (Thermo Fisher Scientific). 0.5 million cells per well were seeded into a 6-well plate, 24 h prior to transfection.

Constructs of the *Tb*CBPs were cloned in a pLIB vector modified with a CMV promoter for mammalian cell expression. The different combinations of plasmids were mixed in an equimolar ratio. 1 μg DNA was diluted in 50 μL with FBS-free DMEM high glucose medium. In parallel, 3 μL of the transfection reagent LipoD293 (SignaGen Laboratories) was diluted in 50 μL of FBS-free DMEM high glucose media. Both dilutions were mixed and incubated for 10 min at room temperature to form the transfection complex. Before transfection, media in each well (80–90 % confluency) was reduced to 1 ml, and the transfection mix was spread equally across the well. After 12–18 h medium was replaced by 2 ml of completed DMEM high glucose medium. Cells were harvested after another 24 h of incubation by centrifugation for 5 min at $500 \times g$. The pellets were frozen at −20 °C until further use.

Cell pellets were resuspended in 1 ml of Buffer A (20 mM HEPES pH 7.5, 150 mM NaCl, 5 mM MgCl₂, 2 % Glycerol) supplemented with a protease inhibitor tablet (cOmplete, EDTA-free-protease inhibitor cocktail) and 2 μg/ml DNase. Subsequently, cells were lysed by sonication (15 s, 30 % Amplitude, 1 s on, 2 s off; Vibra-cell, Sonics). The cell debris was pelleted by centrifugation for 45 min at 4 °C and $21,130 \times g$. Clarified lysate was transferred to a fresh microcentrifuge tube and mixed with 15 μL of pre-equilibrated anti-mCherry resin (CNBr-activated Sepharose (cytiva) coupled with anti-mCherry nanobody). Samples were incubated for 3 h at 4 °C on a steering wheel. After incubation, the resin was washed four times with 500 μL of Buffer A (20 mM HEPES, 150 mM NaCl, 5 mM MgCl₂, 2 % Glycerol), and resuspended in SDS sample loading buffer for 10 min at 95 ˚C. The presence of target proteins was followed up by SDS-PAGE and Western blot.

## Western blot

To follow-up co-precipitation experiments, protein samples were resolved on a stain-free 12 % SDS-PAGE gel together with a pre-stained Page Ruler protein ladder (Thermo Fisher Scientific). SDS-PAGE gels were equilibrated for 10 min in transfer buffer (190 mM Glycine / 20 mM Tris-Base / 20% Ethanol). A PVDF (polyvinylidene difluoride) membrane was activated for 1 min in 100% Ethanol and then equilibrated in transfer buffer for 10 min. A transfer sandwich was assembled, and the transfer was carried out at 80 V for 90 min on ice using a Bio-Rad Mini-Protean Tetra Vertical Electrophoresis Cell on Biorad PowerPac™ Basic. Membranes were blocked using 5 % Bovine Serum Albumin Fraction V (BSA) in 1x PBST (1x PBS / 0.1 % Tween) at 4 °C overnight. The membranes were incubated with anti-mCherry antibody (1:1000, Novus Biologicals 1C51) or anti-HA antibody (1:10000; Sigma-Aldrich H3663) for 1 h at room temperature, washed, and incubated with HRP-conjugated anti-mouse IgG antibody (1:10000, Cell Signaling 7076) for 1 h at room temperature, and washed three times for 10 min with PBST. The membranes were developed using SuperSignal™ West Femto Maximum Sensitivity Substrate imaged using a ChemiDoc MP imager (Bio-Rad).

## RNA Electrophoretic mobility shift assay (RNA-EMSA)

A dilution series of purified protein complexes in protein binding buffer (20 mM HEPES pH 7.5, 50 mM NaCl, 5 mM MgCl₂) was prepared. The different RNAs were diluted with RNA binding buffer (20 mM HEPES pH 7.5, 50 mM NaCl, 5 mM MgCl₂, 0.05 % Tween) to a concentration of 50 nM, mixed 1:1 with protein samples, and incubated for 30 min on ice. Samples were mixed with Orange G loading dye (final

concentration 10 % Glycerol, Orange G) and analyzed on a 6 % non-denaturing polyacrylamide gel (100 V for ~ 90 min at 4 °C). Bands were detected after a 5 min incubation with SybrGold (Invitrogen) using a ChemiDoc MP Imaging System (Bio-Rad). Free RNA bands were quantified relative to the RNA-only control using ImageLab (Bio-Rad), and the dissociation constant was calculated by fitting the data with a Hill model in Prism (GraphPad).

## Saturation transfer difference (STD)
NMR spectroscopy was performed on a Bruker Avance Neo 800 MHz spectrometer equipped with a triple-resonance gradient cryogenic probe. The NMR data was collected and processed using Bruker Top-Spin version 4.1.4. Samples were measured at 298 K and prepared in 20 mM HEPES (pH 7.5), 50 mM NaCl, 5 mM MgCl$_2$, 5% glycerol, and 5% D$_2$O (v/v) added for the lock. For each sample, a reference 1D 1H spectrum using Bruker pulse program *zgesgp* was first acquired with 256 scans, a sweep width of 22.3 ppm (17857 Hz) centered at 4.70 ppm, and the water signal suppressed with excitation sculpting. Saturation transfer difference measurements were first acquired as a pseudo-2D experiment using Bruker pulse program *stddiffesgp.3* with 2 s off-resonance or on-resonance saturation using $^1$H frequencies of − 40 and 10 ppm, respectively. Data were acquired using 32768 points and 1024 scans, with a sweep width of 15.6 ppm centered on 4.70 ppm, and the water signal suppressed by excitation sculpting. The pseudo-2D data was split into two 1D spectra, which were processed with identical parameters, and the difference spectra calculated. Note that the 5 % glycerol required for protein stability prevented the usable detection of many ligand peaks.

## Fluorescence polarization assay
3′-(6FAM) labeled RNA oligomers provided by Biomers (biomers.com) were refolded and diluted in RNA binding buffer (20 mM HEPES pH 7.5, 50 mM NaCl, 5 mM MgCl$_2$, 0.01% Tween) to a final concentration of 50 nM. Purified proteins were buffer-exchanged into protein binding buffer (20 mM HEPES pH 7.5, 50 mM NaCl, 5 mM MgCl$_2$) using Zeba™ Spin Desalting Column with a 7k MWCO (Thermo Fisher Scientific). A serial dilution of the protein samples was prepared in a black 384-well non-binding microplate (Greiner Bio-One) mixed 1:1 with the 50 nM RNA and incubated for 30 min on ice. The polarization value was measured with the CLARIOstar Plus microplate reader (BMG Labtech) with excitation/emission wavelengths of 460/515 nm. The dissociation constant was calculated by fitting the data with a nonlinear sigmoidal fit (4PL) in Prism (GraphPad).

## Microscale thermophoresis (MST)
N-Terminal TAMRA-labeled peptide of *Tb*CBP30 residues R34-G60 was synthesized by SB-PEPTIDE and diluted to 2.71 μM in storage buffer (3 % acetic acid, 30 % acetonitrile), aliquoted and stored at − 80 °C. A fresh aliquot of the peptide was thawed and diluted in peptide binding buffer (20 mM HEPES, 250 mM NaCl, 5 mM MgCl, 0.1 % Tween) to a concentration of 50 nM. The protein sample was thawed, and the buffer was exchanged to protein binding buffer (20 mM HEPES, 250 mM NaCl, 5 mM MgCl) with Zeba™ Spin Desalting Column with a 7 kDa MWCO (Thermo Scientific). The protein was used to prepare a serial dilution in a black 384-well, F-bottom, small volume, non-binding microplate (Greiner bio-one) and mixed 1:1 with *Tb*CBP30 peptide and incubated for 30 min on ice. Samples were measured in triplicate on Monolith™ NT.115 (NanoTemper Technologies) in Monolith™ NT.115 Series Premium Capillaries. We observed a ligand-dependent increase in the initial fluorescence which we used to quantify the binding. An SDS denaturation test (SD-test) following the manufacturer's instructions excluded nonspecific adsorption of the proteins to the capillaries and/or plastic micro reaction tube walls, or effects due to aggregation of the fluorescent molecule upon addition of the ligand (Supplementary Fig. S11). The measurement was done using the green excitation

LED at 100 % Excitation power. The data was evaluated using the Nanotemper MO.Affinity Analysis v2.3 tool.

## Cap4 ligand synthesis
**General Information.** Solvents, chemical reagents, and starting materials were from commercial sources. Commercially available 2′-*O*-methyluridine phosphoramidite was purchased from Biosearch Technologies. Solid support for oligonucleotide synthesis was purchased from GE Healthcare. DNA synthesis grade acetonitrile (< 10 ppm of water) was used for the coupling reaction and for washing the solid support. All work-up and purification procedures were performed with reagent-grade solvents under an ambient atmosphere.

**Ion-exchange chromatography.** The synthesized oligonucleotide was purified by ion exchange chromatography on DEAE Sephadex A-25 (HCO$_3^-$ form). The column was loaded with the reaction mixture and thoroughly washed with deionized water. The products were eluted using a linear gradient of 0–1.2 M triethylammonium bicarbonate (TEAB) in deionized water. The collected fractions were analyzed spectrophotometrically at 260 nm and by RP HPLC. After evaporation to dryness with repeated additions of 96% and 99.8% ethanol, the products were isolated as triethylammonium salts.

**Analytical and preparative chromatography.** Analytical RP HPLC was performed on Agilent Tech. Series 1200 using a Gemini 3 μm NX-C18 LC column (110 Å, 150 × 4.6 mm, 3 μm, flow rate 1.0 ml/min) with a linear gradient elution with 50 mM ammonium acetate buffer, pH 5.9 (buffer A), and 1:1 v/v methanol/buffer A (buffer B) and UV detection at 254 nm: 0–100% in 7.5 min. Semi-preparative RP HPLC was performed on the same instrument using a Gemini 5 μm NX-C18 LC column (110 Å, 150 × 10 mm, flow rate 5.0 ml/min) with a linear gradient of MeCN in 50 mM ammonium acetate buffer (pH 5.9) and UV detection at 254 nm.

**Spectroscopic characteristics.** The structure and purity of the products were confirmed by NMR spectroscopy. NMR spectra were recorded with a Bruker Avance III HD spectrometer at 500.24 MHz ($^1$H NMR), and 202.49 MHz ($^{31}$P NMR) using the 5 mm PABBO BB/19F-1H/D Z-GRD probe at 25 °C. The raw NMR data were processed using MestReNova software. $^1$H NMR chemical shifts were calibrated with HDO signal (4.790 ppm), DMSO-$d_6$ (2.500 ppm), or CDCl$_3$ (7.260 ppm). For the calibration of $^{31}$P NMR, H$_3$PO$_4$ was used as an external standard. The high-resolution mass spectra were recorded on Thermo Scientific LTQ OrbitrapVelos spectrometer.

*Synthesis.* **3′,5′-Diacetyl-2′-*O*-methyladenosine (1):** 2′-*O*-Methyladenosine (5.0 g, 17.8 mmol, 1.0 equiv.) was suspended in a mixture of anhydrous dimethylformamide (14.0 mL) and pyridine (7.0 mL), cooled to 0 °C and acetic anhydride (7.0 mL, 74.8 mmol, 4.2 equiv.) and 4-dimethylaminopyridine (80 mg, 0.65 mmol, 0.04 equiv.) were added. The reaction solution was stirred for 3 h. Methanol (5 mL) was added to quench the reaction. The residual pyridine was removed by coeva-poration with toluene. The product was isolated by flash chromato-graphy (0 → 4% methanol in methylene chloride) to afford compound **1** (4.60 g, 12.6 mmol, 71%) as a white solid (Supplementary Fig. S4A).

**$^1$H NMR (500 MHz, DMSO-$d_6$, 25°C):** δ = 8.38 (s, 1H), 8.16 (s, 1H), 7.36 (s, 2H, NH$_2$), 6.03 (d, $^3J_{H-H}$ = 6.4 Hz, 1H, H1′), 5.50 (dd, $^3J_{H-H}$ = 5.3, 3.2 Hz, 1H, H3′), 4.88 (dd, $^3J_{H-H}$ = 6.5, 5.2 Hz, 1H, H2′), 4.37−4.29 (m, 2H, H4′, H5′), 4.25 (dd, $^2J_{H-H}$ = 11.1, $^3J_{H-H}$ = 5.3 Hz, 1H, H5″), 3.27 (s, 3H, OCH$_3$), 2.14 (s, 3H, CH$_3$), 2.04 (s, 3H, CH$_3$ $_{2′-O}$) ppm.

**9-(3′,5′-Diacetyl-2′-*O*-methylfuranosyl)-6-(1,2,4-triazol-4-yl) purine (2):** 3′,5′-*O*-Diacetyl-2′-*O*-methyladenosine (2.36 g, 6.44 mmol, 1.0 equiv.) *N,N*-bis[(dimethylamino)-methylene]hydrazine dihy-drochloride (3.49 g, 16.8 mmol, 2.6 equiv.) and p-toluenesulfonic acid monohydrate (52.0 mg, 0.27 mmol, 0.02 equiv.) were dissolved in dry toluene and stirred under argon atmosphere in the dark at 110 °C overnight. Then all volatiles were removed under a vacuum. The

residue was dissolved in methylene chloride and washed with 5% citric acid, a saturated solution of sodium bicarbonate, and brine. The organic fractions were combined and the volatiles were removed under vacuum. The product was isolated by flash chromatography (0.5 → 2% methanol in methylene chloride) afford compound **2** (1.07 g, 2.56 mmol, 40%) as a white solid.

**¹H NMR (500 MHz, DMSO-$d_6$, 25 °C):** δ = 9.64 (s, 2H, H$_{triazole}$), 9.04 (s, 1H, H2 or H8), 8.97 (s, 1H, H2 or H8), 6.25 (d, $^3J_{H-H}$ = 6.1 Hz, 1H, H1'), 5.55 (dd, $^3J_{H-H}$ = 5.3 Hz, $^3J_{H-H}$ = 3.7 Hz, 1H, H3'), 4.92 (m, 1H, H2'), 4.43 – 4.34 (m, 2H, H4', H5'), 4.31 (dd, $^2J_{H-H}$ = 11.9, $^3J_{H-H}$ = 5.8 Hz, 1H, H5''), 3.31 (s, 3H, CH$_{3\ 2'-O}$), 2.15 (s, 3H, CH$_3$), 2.05 (s, 3H, CH$_3$) ppm.

**6-(N,N-Dimethyl)-2'-O-methyladenosine (3):** Compound **2** (1.05 g, 2.52 mmol, 1.0 equiv.) was suspended in 33% dimethylamine solution in water (30 mL) and stirred under an argon atmosphere at room temperature for 2 days. During that time, the white suspension turned into a clear solution. All volatiles were removed under vacuum. The crude product was purified by flash chromatography (1 → 3% methanol in methylene chloride) afford compound **3** (857 mg, 2.77 mmol, 91%) as a white solid.

**¹H NMR (500 MHz, CDCl₃, 25 °C):** δ = 8.28 (s, 1H, H2 or H8), 7.74 (s, 1H, H2 or H8), 5.82 (d, $^3J_{H-H}$ = 7.6 Hz, 1H, H1'), 4.78 (dd, $^3J_{H-H}$ = 7.6 Hz, $^3J_{H-H}$ = 4.7 Hz, 1H, H2'), 4.57 (m, 1H, H3'), 4.35 (m, 1H, H4'), 3.96 (dd, $^2J_{H-H}$ = 13.0 Hz, $^3J_{H-H}$ = 1.6 Hz, 1H, H5'), 3.75 (m, 2H, H5''), 3.52 (br s, 6H, 2 × CH$_{3\ N6-Me}$), 3.32 (s, 3H, CH$_{3\ 2'-O}$) ppm

**5'-O-Dimethoxytrityl-6-(N,N-dimethyl)-2'-O-methyladenosine (4):** A mixture of **3** (703 mg, 2.27 mmol, 1.0 equiv.), 4,4'-dimethoxytrityl chloride (2.30 g, 6.81 mmol, 3.0 equiv,) and triethylamine (0.888 mL, 6.37 mmol, 2.8 equiv.) in anhydrous pyridine (12.1 mL) was stirred for 3 h at room temperature. The mixture was quenched with methanol (2.5 mL) and evaporated under reduced pressure at room temperature. The residual syrup was dissolved in dichloromethane (45 mL) and washed with 1 M solution of sodium bicarbonate. The organic layer was dried over anhydrous sodium sulfate, filtered, and evaporated under reduced pressure. The residual pyridine was removed by coevaporation with toluene. The product was isolated by flash chromatography (0 → 50% ethyl acetate in n-hexane with 0.5%$_{v/v}$ TEA) afford compound **4** (958 mg, 1.57 mmol, 69%) as a white solid.

**¹H NMR (500 MHz, CDCl₃, 25 °C):** δ = 8.32 (s, 1H, H2 or H8), 7.82 (s, 1H, H2 or H8), 7.33 – 7.22 (m, 5H, ArH), 7.19 – 7.14 (m, 4H, ArH), 6.85 – 6.80 (m, 4H, ArH), 5.86 (d, $^3J_{H-H}$ = 7.3 Hz, 1H, H1'), 4.73 – 4.66 (m, 1H, H2'), 4.59 (dd, $^3J_{H-H}$ = 4.7 Hz, $^3J_{H-H}$ = 1.1 Hz, 1H, H3'), 4.34 (m, 1H, H4'), 3.95 (dd, $^2J_{H-H}$ = 12.9 Hz, $^3J_{H-H}$ = 1.7 Hz, 1H, H5'), 3.79 (s, 6H, 2 × OCH$_{3\ DMTr}$), 3.77 (m, 1H, H5''), 3.60 (br s, 6H, 2 × CH$_{3\ N6-Me}$), 3.34 (s, 3H, CH$_{3\ 2'-O}$) ppm.

**5'-O-Dimethoxytrityl-6-(N,N-dimethyl)-2'-O-methyladenosine-3'-(2-cyanoethyl)-N,N-diisopropyl-phosphoramidite (5):** Compound **4** (950 mg, 1.55 mmol, 1.0 euiv.) was dissolved in anhydrous methylene chloride (17 mL), and treated with N,N-diisopropylethylamine (1.08 mL, 6.20 mmol, 4.0 equiv.) and 2-cyanoethyl N,N-diisopropylchlorophosphoramidite (0.70 mL, 3.10 mmol, 2.0 equiv.). The reaction solution was stirred under argon atmosphere for 5 h at room temperature. Then the solution was washed with saturated sodium bicarbonate solution. The organic layer was dried over anhydrous sodium sulfate, filtered, and evaporated under reduced pressure. The product was isolated by column chromatography (0 by flash chromatography 50% ethyl acetate in n-hexane with 0.5%$_{v/v}$ TEA) to afford a mixture of diastereomers of **5** (957 mg, 1.18 mmol, 76%) as a white solid.

**Diastereomer 1: ¹H NMR (500 MHz, CDCl₃, 25 °C):** δ = 8.25 (s, 1H, H8), 7.88 (s, 1H, H2), 7.43 – 7.40 (m, 2H, ArH-2,6$_{Ph-DMTr}$), 7.33 – 7.28 (m, 4H, ArH), 7.25 – 7.22 (m, 2H, ArH), 7.21 – 7.17 (m, 1H, ArH), 6.80 – 6.75 (m, 4H, ArH-3,5$_{MeOPh-DMTr}$), 6.11 (d, $^3J_{H,H}$ = 5.8 Hz, 1H, H1'), 4.60 (m, 1H, H3'), 4.55 (m, 1H, H2'), 4.30 (m, 1H, H4'), 3.95 – 3.82 (m, 2H, OCH$_2$CH$_2$CN), 3.76 (s, 6H, 2 × OCH$_{3DMTr}$), 3.60 – 3.47 (m, 8H, 2 × CH$_{3\ N6-Me}$, CH$_{iPr}$, H5'), 3.45 (s, 3H, 2'-O-CH$_3$), 3.29 (m, 1H, H5''), 2.95 (m, 1H, CH$_{iP}$), 2.66 – 2.60 (m, 2H, OCH$_2$CH$_2$CN), 1.16 (d, $^3J_{H,H}$ = 6.5 Hz, 6H,

CH$_{3iPr}$), 1.03 (d, $^3J_{H,H}$ = 6.8 Hz, 6H, CH$_{3iPr}$), **³¹P NMR (202.5 MHz, CDCl₃, 25 °C):** δ = 150.21 (m, 1 P, P) ppm

**Diastereomer 2: ¹H NMR (500 MHz, CDCl₃, 25 °C):** δ = 8.26 (s, 1H, H8), 7.93 (s, 1H, H2), 7.45 – 7.40 (m, 2H, ArH-2,6$_{Ph-DMTr}$), 7.34 – 7.29 (m, 4H, ArH), 7.26 (m, 2H, ArH), 7.23 – 7.17 (m, 1H, ArH), 6.82 – 6.77 (m, 4H ArH-3,5$_{MeOPh-DMTr}$), 6.09 (d, $^3J_{H,H}$ = 5.1 Hz, 1H, H1'), 4.62 (m, 1H, H3'), 4.52 (m, 1H, H2'), 4.35 (m, 1H, H4'), 3.77 (s, 6H, 2 × OCH$_{3DMTr}$), 3.59 – 3.47 (m, 10H, 2 × CH$_{3\ N6-Me}$, OCH$_2$CH$_2$CN, CH$_{iPr}$, H5'), 3.45 (s, 3H, 2'-O-CH$_3$), 3.30 (m, 1H, H5''), 2.98 (m, 1H, CH$_{iP}$), 2.36 (m, 2H, OCH$_2$CH$_2$CN), 1.17 (12H, 4 × CH$_{3iPr}$), **³¹P NMR (202.5 MHz, CDCl₃, 25 °C):** δ = 150.90 (m, 1 P, P) ppm.

**5'-O-Dimethoxytrityl-3-(N-methyl)-2'-O-methyluridine-3'-(2-cyanoethyl)-N,N-diisopropyl-phosphoramidite (6).** Compound **6** was synthesized according to the literature procedure[92]. A 250 mg of 5'-O-DMT-2'-O-methyluridine phosphoramidite (0.329 mmol, 1.0 equiv) and 205 μL of methyl iodide (3.29 mmol, 10 equiv) were dissolved in 3.3 mL of dichloromethane and mixed with 3.3 mL of an aqueous solution of Bu$_4$NBr (0.1 M, 1.0 equiv) and NaOH (1.0 M). The reaction mixture was stirred vigorously 30 min. Then, the reaction mixture was partitioned between water (33 mL) and diethyl ether (133 mL), and the aqueous phase was extracted with ethyl acetate three times (3 × 33 mL). The organic layers were combined, dried over anhydrous sodium sulfate, filtered, and concentrated under reduced pressure. The residue was dissolved in DCM containing 0.5%$_{v/v}$ triethylamine, evaporated using silica gel, and loaded into a solid sample loader. The product was isolated by flash chromatography (0 → 100% ethyl acetate in n-hexane with 0.5%$_{v/v}$ TEA) to afford a mixture of diastereomers of **6** (227 mg, 0.293 mmol, 89%) as a white solid (Supplementary Fig. S4A).

**¹H NMR (500 MHz, CDCl₃, 25 °C):** δ = 8.05 (d, $^3J_{H,H}$ = 8.1 Hz, 1H, H6$_U$), 7.95 (d, $^3J_{H,H}$ = 8.1 Hz, 1H, H6$_U$), 7.41 (m, 2H, ArH$_{Ph-DMTr}$), 7.36 (m, 2H, ArH$_{Ph-DMTr}$), 7.33–7.23 (m, 14H, ArH), 6.84 (m, 8H, 2 × ArH-3,5$_{MeOPh-DMTr}$), 6.04 (d, $^3J_{H,H}$ = 2.7 Hz, 1H, H1'), 5.99 (d, $^3J_{H,H}$ = 1.8 Hz, 1H, H1'), 5.32 (d, $^3J_{H,H}$ = 8.1 Hz, 1H, H5$_U$), 5.29 (d, $^3J_{H,H}$ = 8.1 Hz, 1H, H5$_U$), 4.61 (m, 1H, H3'), 4.46 (m, 1H, H3'), 4.24 (m, 1H, H4'), 4.21 (m, 1H, H4'), 3.92 (m, 1H, H2'), 3.88 (m, 2H, H2', OCH$_2$CH$_2$CN), 3.83 (m, 1H, OCH$_2$CH$_2$CN), 3.80 (s, 3H, OCH$_{3\ DMTr}$), 3.80 (s, 3H, OCH$_{3\ DMTr}$), 3.79 (s, 3H, OCH$_{3\ DMTr}$), 3.79 (s, 3H, OCH$_{3\ DMTr}$), 3.68–3.41 (m, 10H, OCH$_2$CH$_2$CN, 2 × H5', 2 × H5', 4 × CH$_{iPr}$), 3.60 (s, 3H, CH$_{3\ N3-Me}$), 3.60 (s, 3H, CH$_{3\ N3-Me}$), 3.32 (s, 6H, 2 × CH$_{3\ 2'-O}$), 2.64 (m, 2H, OCH$_2$CH$_2$CN), 2.40 (t, $^3J_{H,H}$ = 6.2 Hz, 2H, OCH$_2$CH$_2$CN), 1.19 (d, $^3J_{H,H}$ = 6.7 Hz, 6H, CH$_{3\ iPr}$), 1.19 (d, $^3J_{H,H}$ = 6.7 Hz, 6H, CH$_{3\ iPr}$), 1.16 (d, $^3J_{H,H}$ = 6.8 Hz, 6H, CH$_{3\ iPr}$), 1.03 (d, $^3J_{H,H}$ = 6.8 Hz, 6H, CH$_{3\ iPr}$) ppm; HRMS (ESI) m/z: [M + H]$^+$ calcd for C$_{41}$H$_{52}$N$_4$O$_9$P$^+$ 775.34664, found: 775.34746.

**p$^{m6,6}$A$_m$pA$_m$pC$_m$p$^{m3}$U$_m$pA.** Oligonucleotide p$^{m6,6}$A$_m$pA$_m$pC$_m$p$^{m3}$U$_m$pA was synthesized as reported previously[92]. Solid-phase syntheses of short oligonucleotides were performed in a 10 mL syringe equipped with frit and loaded with polystyrene support [ribo A 300 PrimerSupport 5 G (299 μmol/g, GE Healthcare)]. The synthesis scale was 50 μmol (based on the support loading provided by the manufacturer). The detritylation step was performed by passing 40 mL of 3% $_{(v/v)}$ trichloroacetic acid in DCM through the column. The solid support was washed with 40 mL of DNA synthesis grade acetonitrile (< 10 ppm of H$_2$O) and dried in a vacuum desiccator. In the coupling step, a 0.3 M solution of an appropriate phosphoramidite (2.0 equivalents) in anhydrous acetonitrile and a 1.5 volume of 0.3 M BTT Activator were shaken with the support for 30 min. Then, the support was washed with 40 mL of acetonitrile and the phosphite triester was oxidized by passing 15 mL of 0.05 M iodine in pyridine/water 9:1 $_{v/v}$. The cycle was repeated, once for each base, to produce the final oligonucleotide. To prepare the oligonucleotide 5'-phosphates, the bis(2-cyanoethyl)-N,N-diisopropylphosphoramidite (3.0 equivalents, 0.3 M in acetonitrile + 1.5 volume of 0.3 M BTT Activator) was used in the last cycle, and the detritylation step was omitted. After the last cycle of the synthesis, 2-cyanoethyl groups were removed by passing 20 mL of 20%$_{v/v}$ solution of diethylamine in acetonitrile. The

support was dried in a vacuum desiccator and transferred to a 50 mL polypropylene tube, and the oligonucleotide was cleaved from the support using AMA (1 mL, 1:1 $_{v/v}$ mixture of 33% ammonium hydroxide and 40% methylamine in water for 3 h at 37 °C (Eppendorf ThermoMixer C, 1000 rpm). The suspension was filtered, washed with water, evaporated to dryness, redissolved in water, and freeze-dried. The residue was dissolved in 200 μL of DMSO, followed by the addition of triethylamine (430 μL) and triethylammonium trihydrofluoride (TEA·3HF, 250 μL), and the resulting mixture was shaken for 3 h at 65 °C (Eppendorf Thermo-Mixer C, 1000 rpm). The reaction was quenched by the addition of 0.05 M NaHCO₃ in water (ca. 20 mL). The product was isolated by ion-exchange chromatography on DEAE Sephadex using a linear gradient of TEAB (0–1.2 M), evaporated to dryness with ethanol to give a white solid (206 mOD, 3.3 μmol, 7%) (Supplementary Fig. S4B).

*Cap4*. Cap4 was synthesized as reported previously[92]. Triethylammonium salt of p$^{m6,6}$A$_m$pA$_m$pC$_m$p$^{m3}$U$_m$pA (206 mOD, 3.3 μmol) was dissolved in anhydrous DMF (132 μL) followed by the addition of imidazole (14.4 mg, 211 μmol), triethylamine (11 μL, 79 μmol), 2,2′-dithio-dipyridine (17.4 mg, 79 μmol), and triphenylphospine (20.7 mg, 79 μmol). After 5 h, the product was precipitated with a cold solution of NaClO₄ (16.2 mg, 132 μmol) in acetonitrile (1.32 mL). The precipitate was centrifuged (4700 × *g*, 6 min) in a 50 mL conical tube at 4 °C, washed with cold acetonitrile by centrifugation 3 times, and dried under reduced pressure. Thus obtained *P*-imidazolide was mixed with 7-methylguanosine 5′-diphosphate (30 mg, 33.0 μmol) in anhydrous DMSO (440 μL), followed by the addition of anhydrous ZnCl₂ (72 mg, 528 μmol). The mixture was stirred for ca. 14 h, and the reaction was quenched by the addition of 8.5 mL of aqueous solution of EDTA (20 mg/mL) and NaHCO₃ (10 mg/mL). The product was isolated by ion-exchange chromatography on DEAE Sephadex using a linear gradient of TEAB (0–1.2 M) and purified by semi-preparative RP HPLC (gradient elution 0–15% acetonitrile in 0.05 M ammonium acetate buffer pH 5.9) to afford—after evaporation and repeated freeze-drying from water—ammonium salt of **27** m⁷Gppp$^{m6,6}$A$_m$pA$_m$pCp$^{m3}$U$_m$pA (5.67 mg, 115 mOD, 1.88 μmol, 57%) as a white amorphous solid. HRMS (ESI) *m/z*: [M + H]⁺ calcd for $C_{66}H_{89}N_{25}O_{44}P_7^-$ 2152.36640, found: 2152.36410 (Supplementary Fig. S4C, D).

## Cap4-SLRNA synthesis

**RNA solid-phase synthesis.** RNA synthesis was performed on an H6 GeneWorld automated DNA/RNA synthesizer (K&A, Laborgeraete GbR, Germany) at a 1.0 μmol scale using a standard phosphoramidite chemistry. 2′-*O*-TOM standard RNA nucleoside building blocks, 2′-*O*-methyl adenosine and 2′-*O*-methyl cytidine phosphoramidite, and 2′-*O*-TBS 1000 Å CPG solid supports were purchased from ChemGenes. Phosphoramidites of m³Um and m⁶₂Am were synthesized according to the described procedure[93]. Detritylation, coupling, capping and oxidation reagents were dichloroacetic acid/1,2-dichloroethane (4/96), phosphoramidite/acetonitrile (100 mM) and benzylthiotetrazole/acetonitrile (300 mM), Cap A/Cap B (1/1) (Cap A: 4-(dimethylamino) pyridine/acetonitrile (500 mM), Cap B: acetic anhydride/sym-collidine/acetonitrile (2/3/5)) and iodine (20 mM) in tetrahydrofuran/pyridine/H₂O (35/10/5), respectively. Solutions of phosphoramidites and tetrazole were dried over activated molecular sieves (3 Å) overnight.

**Deprotection, purification, and quantification of RNAs.** The solid-supported RNAs were treated with 1,8-diazabicycloundec-7-en (DBU) in acetonitrile (1.0 M, 0.5 mL) for 5 min at room temperature, then washed with acetonitrile and dried. The CPG beads were transferred to a screw-capped vial and incubated with a mixture of aqueous methylamine (40%, 0.5 mL) and aqueous ammonia (28%, 0.5 mL) for 2 h at 40 °C. The supernatant was removed, and the solid support was washed three times with H₂O/THF (1.0 mL; 1/1). The combined supernatant and washings were evaporated to dryness, and the residue was dissolved in a solution of tetrabutylammonium fluoride in tetrahydrofuran (1.0 M,

1.5 mL) to remove the 2′-*O*-silyl protecting groups. After incubation at 37 °C for 16 h, the reaction was quenched by the addition of triethylammonium acetate/H₂O (1.0 M, 1.5 mL, pH 7.4). Tetrahydrofuran was removed under reduced pressure, and the sample was desalted by size-exclusion column chromatography (GE Healthcare, HiPrep™ 26/10 Desalting; Sephadex G25) eluting with H₂O; collected fractions were evaporated and the RNA was dissolved in H₂O (1 mL).

Crude RNA was purified by anion exchange chromatography (Thermo Scientific Ultimate 3000 HPLC System) on a semi-preparative Dionex DNAPac® PA-100 column (9 mm × 250 mm) at 80 °C with a flow rate of 2 mL/min (eluent A: 20 mM NaClO₄ and 25 mM Tris·HCl (pH 8.0) in 20% aqueous acetonitrile; eluent B: 0.6 M NaClO₄ and 25 mM Tris·HCl (pH 8.0) in 20% aqueous acetonitrile). Fractions containing RNA were evaporated and the residue redissolved in 0.1 M triethylammonium bicarbonate solution (10 to 20 mL), loaded on a C18 SepPak Plus® cartridge (Waters/Millipore), washed with H₂O, and then eluted with acetonitrile/H₂O (1/1).

Crude and purified RNA were analyzed by anion exchange chromatography (Thermo Scientific Ultimate 3000 HPLC System) on a Dionex DNAPac® PA-100 column (4 mm × 250 mm) at 80 °C with a flow rate of 1 mL/min. A gradient of 0–45% B in 60 min was applied; eluent A: 20 mM NaClO₄ and 25 mM Tris·HCl (pH 8.0) in 20% aqueous acetonitrile; eluent B: 0.6 M NaClO₄ and 25 mM Tris·HCl (pH 8.0) in 20% aqueous acetonitrile. HPLC traces were recorded at UV absorption by 260 nm. RNA quantification was performed on an Implen P300 Nanophotometer.

**Mass spectrometry of oligoribonucleotides.** RNA samples (*ca.* 200 pmol) were diluted with an aqueous solution of ethylenediaminetetraacetic acid disodium salt dihydrate (Na₂H₂EDTA) (40 mM, 15 μL). Water was added to obtain a total volume of 30 μL. The sample was injected onto a C18 XBridge column (2.5 μm, 2.1 mm × 50 mm) at a flow rate of 0.1 mL/min and eluted using gradient 0 to 100% B at 30 °C (eluent A: 8.6 mM triethylamine, 100 mM 1,1,3,3,3-hexafluoroisopropanol in H₂O; eluent B: methanol). RNA was detected by a Finnigan LCQ Advantage Max electrospray ionization mass spectrometer with 4.0 kV spray voltage in negative mode.

**Preparation of capped RNAs (40 nt).** The synthesis of capped RNAs was performed according to the published protocol[93] which was improved and included the following steps: chemical solid-phase synthesis of short Gppp-RNAs (Supplementary Fig. S9, steps 1–5), enzymatic N7 methylation (Supplementary Fig. S9, step 6) and enzymatic ligation (Supplementary Fig. S9, step 7).

*Synthesis of short Gppp-RNAs on solid support (steps 1-5*, Supplementary Fig. S9):

For steps 1–4, the CPG support, placed in the synthesis cartridge, was treated with the appropriate reagent and allowed to shake at room temperature for the indicated time. For phosphitylation, oxidation and Gpp attachment, a few beads of activated 3 Å molecular sieve were added to the syringe.

Steps 1–4: The fully protected resin-bound 5′-OH RNA (1.0 μmol) was treated with 0.5 mL of phosphitylation solution (0.5 mL of diphenyl phosphite, 2.0 mL of anhydrous pyridine), which was manually passed through the column (using plastic syringes) and allowed to react for 10 min. After washing the beads with ACN, 0.5 mL of hydrolysis solution (0.5 mL of 1 M aqueous triethylammonium bicarbonate, 2.5 mL of H₂O, and 2.0 mL of ACN) was applied for 20 min. The solid support was then washed with ACN and dried *in vacuo* for 2 h. Then, the CPG beads were treated with 0.5 mL of oxidation solution (300 mg of imidazole, 1.0 mL of *N,O*-bis(trimethylsilyl)acetamide, 2.0 mL of anhydrous acetonitrile, 2.0 mL of bromotrichloromethane and 0.2 mL of triethylamine) for 1 h. The solution was removed, and the support was washed with ACN. Finally, Gpp attachment was achieved by applying 0.5 mL of coupling solution (0.28 M guanosine 5′-

diphosphate di-tributylammonium salt in dry DMF and 500 mM of zinc chloride) for 17 h. The solution was removed, and the support was washed with $H_2O$ ($2 \times 0.5$ mL), 40 mM aqueous solution of EDTA ($2 \times 0.5$ mL), and ACN ($3 \times 0.5$ mL) and then dried *in vacuo*.

Step 5: The short capped RNA was deprotected and purified by AE HPLC (for conditions see section: Deprotection, purification and quantification of RNAs). The final product was confirmed by LC-ESI mass spectrometry. Yields and MS data are shown in Supplementary Table S5.

*Enzymatic N7 methylation of Gppp-RNA (step 6, Supplementary Fig. S9):*

Lyophilized Gppp-RNA (30 nmol, 1.0 equiv) was dissolved in a buffer (60 µL, 1.5 mM NaCl, 200 mM $Na_2HPO_4$, pH 7.4) followed by the addition of an aqueous solution of *S*-adenosylmethionine (135 µL, 135 nmol, 4.5 equiv), and the addition of water to obtain a total volume of 516 µL. In the mixed solution, 12 µL of 50 µM MTAN (5′-methylthioadenosine/*S*-adenosylhomocysteine nucleosidase), 12 µL of 50 µM LuxS (*S*-ribosyl homocysteine lyase) and 60 µL of 50 µM Ecm1 (*Encephalitozoon cuniculi* mRNA cap (guanine N7) methyltransferase) were added sequentially; enzymes were provided by the group of Andrea Rentmeister, University of Munich, Germany. After incubation at 37 °C for 1 h, the reaction mixture was stopped by phenol/chloroform extraction. Analysis of the reaction and purification of the product were performed by anion exchange chromatography. The final product was confirmed by LC-ESI mass spectrometry. Yields and MS data are shown in Supplementary Table S5.

*Enzymatic ligation of m⁷Gppp-RNAs (step 7, Supplementary Fig. S9):*

The 40 nt capped RNA was prepared by splinted enzymatic ligation using T4 DNA ligase (*Thermo Scientific*). The 18 nt capped RNA (10 µmol, 1.0 equiv), a chemically synthesized 22 nt 5′-p-RNA (12.5 µmol, 1.25 equiv), and a 23 nt DNA splint (12.5 µmol, 1.25 equiv, *Sigma-Aldrich*) were combined in a final volume of 700 µL and heated at 70 °C for 2 min. The solution was then passively cooled to room temperature for 10 min. For the ligation reaction, 10 × ligation buffer (100 µL, *Thermo Scientific*), PEG (100 µL, *Thermo Scientific*), and T4 DNA ligase (100 µL, 5U/µL) were added and incubated for 3 h at 37 °C. The ligation reaction was stopped by phenol/chloroform extraction. Analysis of the reaction and purification of the ligation product were performed by anion exchange chromatography and the final product was confirmed by LC-ESI mass spectrometry. Yields and MS data are shown in Supplementary Table S5.

### Reporting summary

Further information on research design is available in the Nature Portfolio Reporting Summary linked to this article.

## Data availability

The coordinates and cryo-EM maps were deposited in the PDB and EMDB database: *Tb*CBP20-*Tb*CBP110-*Tb*CBP30-*Tb*CBP66 tetramer cap0 PDB:9F3F and EMD-50173; *Tb*CBP20-*Tb*CBP110-*Tb*CBP30 trimer cap4 PDB:9F67 and EMD-50217. Source data are provided in this paper.

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

## Acknowledgements

We acknowledge Martin Pelosse for support in using the Eukaryotic Expression Facility at EMBL Grenoble, and Sarah Schneider for support in using the EM Facility at EMBL Grenoble. This work benefited from access to the cryo-EM platform of the Structural and Computational Biology Unit at EMBL Heidelberg and CM01 at the European Synchrotron Radiation Facility (https://doi.org/10.15151/ESRF-ES-624205223), and we thank Félix Weis and Grégory Effantin for their assistance with the cryo-EM data collection. We thank the staff at ESRF beamline ID30A-1 for assistance with the SAXS measurements. R.M. and K.B. thank Ann-Marie Lawrence-Dörner (University of Münster) and Andrea Rentmeister (LMU) for a generous gift of Ecm1 (*Encephalitozoon cuniculi* mRNA cap (guanine N7) methyltransferase), MTAN (5′-methylthioadenosine/*S*-adenosylhomocysteine nucleosidase), and LuxS (*S*-ribosyl homocysteine lyase). We thank Caroline Mas for training and assistance with mass photometry, MALLS, and MST measurements. This work used the platforms of the Grenoble Instruct-ERIC Center (ISBG; UAR 3518 CNRS-CEA-UGA-EMBL) within the Grenoble Partnership for Structural Biology (PSB), supported by FRISBI (ANR-10-INBS-0005-02) and GRAL, financed within the University Grenoble Alpes graduate school (Écoles Universitaires de Recherche) CBH-EUR-GS (ANR-17-EURE-0003). We thank the structural biology facility at the European Institute of Chemistry and Biology (CNRS UAR 3033, Inserm US001) for access to the NMR spectrometers. Funding was provided by the Austrian Science Fund FWF (P31691 and F8011-B to R.M.), the Tyrolean Science Fund TWF (F.33309/2021 to K.B.), and the Austrian Research Promotion Agency FFG (858017 to R.M.). Financial support was given from the National Science Center, Poland (2019/33/B/ST4/01843 to J.J.). The lab of E.K. is supported through EMBL core funding and a grant from the French Agence Nationale de la Recherche (ANR-20-CE11-0016 to E.K.). We thank Life Science Editors for editing services (www.lifescienceeditors.com). The authors thank all Kowalinski, Jemiely, and Micura lab members for their discussions and comments throughout the course of the project.

## Author contributions

E.K. conceptualized and led the study. B.P. cloned the complexes and performed an initial purification. H.B., H.P., L.T., and H.G. conducted protein purifications and biochemical assays. H.B. conducted cryo-EM sample preparation, data collection, data processing, and model building, assisted by L.G.D. K.Z. and M.W. conducted cap4 hexanucleotide synthesis supervised by J.J. K.B. conducted cap0 and cap4-oligonucleotide synthesis supervised by R.M. C.M. collected and analyzed the STD-NMR experiment. E.K. and H.B. analyzed the data. E.K. wrote the manuscript, assisted by H.B. and L.G.D.

## Funding

## Competing interests

The authors declare no competing interests.
