## [Peer Review file · Nature Communications]

Structural basis of Spliced Leader RNA recognition by the *Trypanosoma brucei* cap-binding complex

Corresponding Author: Dr Eva Kowalinski

Version 0:

Reviewer comments:

Reviewer #1

(Remarks to the Author)

The manuscript by Bernhard et al presents their determination of the structure of the core of the cap binding complex of *T. brucei*, together with their biochemical validation of their structures and their mode of RNA binding. This is a high-quality piece of work, addressing an important question related to the unusual RNA processing capability of the African trypanosomes.

The structure appears well done from the manuscript and data table, but I had a few concerns when I looked at the models and maps. I first looked at _9085625_ and _9085626_, which appear to be a pair. The protein density and model look very nice, but the density for the ligand (the 7MG) is actually rather weak, with a sigma of 0.5 required to see much. It would be important to have some validation that this density is the 7MG molecule. Either an apo structure, lacking the ligand or some mutagenesis of residues which contact the ligand. The other structure (_9085627_ and _9085628_) was harder to interpret as the model for the ligand appears broken in its connectivity and so looks very strange when opened in coot. These issues should be addressed and are important.

I would recommend some change in the way in which the binding data is described, in particular for the OH-SLe RNA. In binding to the trimer and dimer, the authors state that these '...do not interact with OH-SLe'. (Figure 3FG) while for the tetramer in Figure 4A, then say that it does bind. There is clearly a large change in affinity, but there is also some binding in Figure 3FG. I would recommend to move away from the yes/no interpretation to one about the quantitative change in affinity.

There are also some issues with the pull down data in Figure 4C and 4F. In both cases, it is hard to understand how this pull down was done. Which component was used for pull-down? What is shown in the blot and why? How many times was this repeated and was it quantified?

Finally, I think that the citations for Figure 4E and F are the wrong way round in the main text.

In other very minor comments:

For the SI file, please put each legend on the same page as the relevant figure.

Defocus in the table – do you mean -2.4 not 2.4?

Tryptophane is used occasionally.

Reviewer #2

(Remarks to the Author)

In their manuscript 'Structural basis of Spliced Leader RNA recognition by the *Trypanosoma brucei* cap-binding complex' Bernhard et al describe the cryoEM structure of the trypanosome cap-binding complex and define interactions with a broad range of synthetic RNA constructs. Using an extensive array of different molecular and structural biology techniques they show that the CBP66 subunit, which cryoEM density is absent, is flexibly tethered to the m7G-cap binding CBP20-CBP110

core complex via CBP30 and interacts with double-stranded SL RNA features.

This is a landmark study that sheds light on the trypanosome cap-binding complex. It is of interest for a broad readership and will inform a plethora of future functional studies on trypanosome central mRNA metabolism. To my opinion the article would be well placed in Nature Comm.

The experimental approaches are technically sound and results well and appropriately reported.

I have just a few very minor comments for improving this excellent article.

Page 15/16 wrong Figure references, the MST data is shown in Figure 4E; TbCBP66 coIP in Figure 4F

Line 334-337

The affinity of a TAMRA-labelled TbCBP30 peptide comprising residues R34-G60 towards the TbCBP20-TbCBP110 dimer was determined to $980 \text{ nM} \pm 386 \text{ nM}$, by microscale thermophoresis (Figure 4E, S10).

Line 366-367

Our data reveal that a region within the C-terminus of TbCBP30 (residues K150 to A200) is required for the co-purification of full-length TbCBP66 (Figure 4F).

Line 473-475

These observations indicate high stability of the complex and would support that tetrameric CBC forms a permanent assembly.

Line 483-485

The RNA processing machinery of *T. brucei* is the target of the drug acoziborole (SCYX-7158), which is effective against human African trypanosomiasis (HAT), demonstrating that drugging the RNA metabolism of trypanosomes is an effective strategy.

Line 521

The abbreviation for tris(2-carboxyethyl)phosphine should be introduced and changed to the common 'TCEP'
Same for PMSF

Reviewer #3

(Remarks to the Author)

Bernhard et al report a cryo-EM structural analysis of the *Trypanosoma brucei* cap binding complex (CBC) that is composed of 4 proteins, CBP20/110/30/66, bound to the m7G cap and/or SL RNA. In their cryo-EM focused study, the authors tried to resolve the dimer/trimer/tetramer structures of these proteins to overcome the partial/localized structural ambiguity and understand structural mechanism in the assembly of cap-binding complex. Due to the flexibility in the region of CBP66 and partial CBP30, the authors could not obtain high resolution structures for these two proteins in the complex. SAXS analysis supports the presence of disordered region and the associations of these proteins to the core CBP20/110 complex, where the CBP30 tethers CBP66, like a 'bridge', is supported by mass photometry and size exclusion. By investigating three oligomeric forms, key contact regions are identified, especially CBP30 and 66 to the core complex, and further supported by mutagenesis, MST, CoIP, SEC and EMSA. The authors also confirmed that CBP110 resembles human CBP80, based on structural homology. They recognition of m7GMP in the RNA 5'-cap by CBP20 is revealed. Nevertheless, the final structure remains undefined for partial CBP30 and the whole CBP66.

Overall, this is interesting and relevant work, that provides significant novel insight into an important and kinetoplastid-specific aspects of gene expression, which may be exploited for therapeutic approaches eventually. The data are technically sound, and structural findings are combined with biochemical and biophysical methods. What is missing is to link the structural data with biological functional, e.g. confirming the functional significance of the observed interactions in cellular assays. This would significantly strengthen the impact of the study, but would of course also require significant experimental work.

The authors should consider the following points in a revised manuscript:

Specific comments:

- Could the authors support the functional significance for some of the kinetoplastid unique structural features by cellular experiments?
- It is unfortunate that important structural details are not resolved due to flexibility and thus their contribution to RNA binding are not well resolved. Have the authors considered to provide additional (structural) information for the RNA recognition by RRM and Zn fingers from crystallography or NMR?
- EMSA was used to characterize SL RNA binding with and w/o cap (Fig. 3 F, G) the data must be shown at least in a supplementary material and fitted parameters including Hill factors provided in a table.
- Fig. 5E: what are the Hill factors, can they be rationalized with the structure?
- Why not using a more precise method to quantify the affinity, e.g. ITC, or fluorescence spectroscopy?

Version 1:

Reviewer comments:

Reviewer #1

(Remarks to the Author)

I am largely content with the changes made to the manuscript in response to my comments.

They have presented improved data, including SD-NMR, which indicates specific binding, which appears of good quality. My question was more related to whether the electron density for the ligand was sufficiently high quality to unambiguously show that this is the ligand and the authors have not really added more data on this point. I had imagined mutagenesis of the binding pocket and binding assays being included in a revision. However, the density shown in Figure 1G looks convincing and I accept the authors point that mutagenesis has been done for homoxglues and so it is highly likely that this is the right attribution of density.

For point 2, the authors have made a change, but I had imagined that they would report the fold difference in affinity, rather than using a word such as 'strong' which is a value judgement.

Otherwise I am content.

Reviewer #3

(Remarks to the Author)

The authors have responded to my previous comments and provide additional experiments (although no experimental structural information for the missing, but important region). They also provided additional experiments in response to other reviewers.

I recommend publication.

Point-by-point response to reviewers' comments

The authors thank the reviewers for their feedback and recognition of the impact of our study. We are grateful for your intellectual contributions leading to the improvement of our manuscript. Thank you for your time to evaluate and consolidate our work.

Eva Kowalinski

In the name of all authors.

Reviewer #1 (Remarks to the Author):

The manuscript by Bernhard et al presents their determination of the structure of the core of the cap binding complex of *T. brucei*, together with their biochemical validation of their structures and their mode of RNA binding. This is a high-quality piece of work, addressing an important question related to the unusual RNA processing capability of the African trypanosomes.

1.

The structure appears well done from the manuscript and data table, but I had a few concerns when I looked at the models and maps. I first looked at _9085625_ and _9085626_, which appear to be a pair. The protein density and model look very nice, but the density for the ligand (the 7MG) is actually rather weak, with a sigma of 0.5 required to see much. It would be important to have some validation that this density is the 7MG molecule. Either an apo structure, lacking the ligand or some mutagenesis of residues which contact the ligand. The other structure (_9085627_ and _9085628_) was harder to interpret as the model for the ligand appears broken in its connectivity and so looks very strange when opened in coot. These issues should be addressed and are important.

Consistent with the display parameters for all DeepEMhancer post-processed maps in the paper, Figure 1G, shows the ligand at a map level of 0.03, revealing density for the m⁷GMP portion of the added ligand m⁷GpppA.

To validate the ligand specificity of *Tb*CBC for m⁷G, we performed **additional competition pull-down experiments with different GTP analogues** (replacement of Figure 1D, the original panel has been moved to Figure S1C) and **NMR analysis that we added** as Supplementary Figure S6. In the initial NMR approaches to assay the binding of m⁷GTP and other nucleotides to the *Tb*CBC-trimer, we aimed to use titrations to follow free and bound ligands. The strategy was to follow the ³¹P signal of the nucleotides (or in one case the ¹⁹F signal of m⁷GMP- α -F) in the free and bound forms. Due to the tendency of the *Tb*CBC-trimer to aggregate, and the required long acquisition times, we observed significant protein precipitation. We were unable to find a compromise with the necessary protein concentration and sufficient signal-to-noise

in the NMR spectrum. As a result, we changed our approach to a technique that - although more qualitative than quantitative - required a lower protein concentration and a signal-to-noise ratio related to the free nucleotide. Based on the acquisition of saturation transfer difference (STD) spectra we were able to demonstrate an interaction between the *Tb*CBC-trimer and m⁷GTP, with lower binding to GTP and little or no interaction with ATP. This binding preference remains with a mix of nucleotide ligands, and a final addition of the cap4 RNA shows that the binding of all ligands share the same site on the *Tb*CBC-trimer. This experiment is now referenced in the main text and appears in Supplementary Figure S6.

Text changes to include the NMR experiment:

Initial:

"The conservation of these residues, which serve as determinants for guanine selectivity in human CBP20, suggests the same preference for the *T. brucei* protein (Mazza et al. 2001). This is in line with data showing that m⁷GTP competed efficiently with cap4-SL RNA while ^{2,2,7}GpppG and ApppG were inefficient (Li and Tschudi 2005)."

Rephrased:

"Our nucleotide-competition pull-down assay (Figure 1D) and nuclear magnetic resonance spectroscopy data of a Saturated transfer difference (STD) experiment (Figure S6) indicate the specificity of *Tb*CBC for m⁷G compared to GTP and ATP, consistent with the preference of the human protein (Mazza et al. 2001). This is in line with data showing that m⁷GTP competed efficiently with cap4-SL RNA binding to *Tb*CBC, while ^{2,2,7}GpppG and ApppG were inefficient competitors (Figure 1D, Li and Tschudi 2005)."

Extensive mutational analysis of the cap-binding residues of CBP20 has been performed by Worch et al. 2008 and also Mazza et al. 2002 (table 2 in their paper). The assayed cap-binding residues are conserved between *Tb*CBC and human CBC (compare Figure 3B,C and the alignment in Figure S8). In general, all our cryo-EM samples of *Tb*CBC were greatly stabilized by the addition of cap-analogues, judged by the improved quality of cryo-grids. With the revision, we re-assessed the *Tb*CBC-tetramer dataset and processed previously discarded classes that lacked coulomb density in the region of CBP20 with the aim of providing a reconstruction of a ligand-free *Tb*CBC. Despite the anisotropy of the apo-map, the map (3 Å overall resolution) confirms the intrinsically unfolded behaviour of *Tb*CBC20 in the absence of ligand, consistent with the human complex where N-and C-terminal extensions of the central RNP cooperatively adopt their secondary structure in the presence of the cap analogue. We added panels to Figure S2 and modified the main text:

Initial:

"The sequence conservation of *Tb*CBP20 initially led to the identification of the *T. brucei* CBC (Li and Tschudi 2005). Our data reveal the *Tb*CBP20 structure with a central RNP motif forming a β -sheet and additional N- and C-terminal domains is highly similar to its human homologue. In human CBP20, the N- and C-terminal domains are intrinsically unfolded and adopt a secondary structure only upon binding to the RNA cap (Mazza et al. 2002; Worch et al. 2009). This might be similar in the *T. brucei* complex since during data processing of the CBC reconstruction, we eliminated particles with an incomplete *Tb*CBP20 N-terminal domain to increase particle homogeneity. These particles might be those that are not bound to a cap analogue (Figure S2). The C-terminal domain of *Tb*CBP20 is unresolved in the *Tb*CBC-tetramer.

Rephrased:

The sequence conservation of *Tb*CBP20 initially led to the identification of the *T. brucei* CBC (Li and Tschudi 2005). Our data reveal the *Tb*CBP20 structure with a central RNP motif forming a β -sheet and additional N- and C-terminal domains is highly similar to its human homologue. The C-terminal domain of *Tb*CBP20 is unresolved in our maps. In human CBP20, parts of the N- and C-terminal domains are intrinsically unfolded and adopt a secondary structure only upon binding to the RNA cap (Mazza et al. 2002; Worch et al. 2009). A map reconstruction from discarded, incomplete particles of the *Tb*CBC-tetramer dataset, probably deficient of the m⁷G ligand, is highly anisotropic, but reveals a similar intrinsically unfolded behaviour of the N- and C-termini of *Tb*CBP20, pointing to a conserved induced fit m⁷G binding mechanism (Figure S2)."

The authors apologise for potential issues with the original trimer-cap4 structure file. With the revision, we provide the final pdb deposited cif file in which the library issues are fixed.

2.

I would recommend some change in the way in which the binding data is described, in particular for the OH-SLe RNA. In binding to the trimer and dimer, the authors state that these '...do not interact with OH-SLe'. (Figure 3FG) while for the tetramer in Figure 4A, then say that it does bind. There is clearly a large change in affinity, but there is also some binding in Figure 3FG. I would recommend to move away from the yes/no interpretation to one about the quantitative change in affinity.

The text has been adapted

Initial: The *Tb*CBC dimer and trimer do not interact with OH-SLe, indicating that the presence of the m⁷G-cap is a strict requirement for the interaction of the SL RNA with the *Tb*CBP20-*Tb*CBP110 core complex

Rephrased:

The *TbCBC* dimer and trimer do not strongly interact with OH-SLe, indicating that the presence of the m⁷G-cap is a requirement for the interaction of the SL RNA with the *TbCBP20-TbCBP110* core complex

3.

There are also some issues with the pull down data in Figure 4C and 4F. in both cases, it is hard to understand how this pull down was done. Which component was used for pull-down? What is shown in the blot and why? How many times was this repeated and was it quantified?

We add a sentence to improve clarity:

We used transient overexpression of HA-tagged *TbCBC* subunits in HEK293T cells followed by pull-down via mcherry-*TbCBP30* as bait to assess complex formation with mutant protein."

And later include "affinity-tagged" to improve clarity.

For correctness "co-immuno-precipitation" was replaced in all instances by "co-precipitation" since we do not use antibodies which "immuno-" would imply.

We indicate bait and prey in the respective Figure panels.

To indicate reproducibility, we added to the figure legends: "A representative blot of more than 3 experiments is shown "

4.

Finally, I think that the citations for Figure 4E and F are the wrong way round in the main text.

The figure citations were corrected.

In other very minor comments:

5.

For the SI file, please put each legend on the same page as the relevant figure.

Rearrangement done.

6.

Defocus in the table – do you mean -2.4 not 2.4?

Mistake corrected.

7.

Tryptophane is used occasionally.

The typos in tryptophan have been corrected.

Reviewer #2 (Remarks to the Author):

In their manuscript 'Structural basis of Spliced Leader RNA recognition by the Trypanosoma brucei cap-binding complex' Bernhard et al describe the cryoEM structure of the trypanosome cap-binding complex and define interactions with a broad range of synthetic RNA constructs. Using an extensive array of different molecular and structural biology techniques they show that the CBP66 subunit, which cryoEM density is absent, is flexibly tethered to the m7G-cap binding CBP20-CBP110 core complex via CBP30 and interacts with double-stranded SL RNA features.

This is a landmark study that sheds light on the trypanosome cap-binding complex. It is of interest for a broad readership and will inform a plethora of future functional studies on trypanosome central mRNA metabolism. To my opinion the article would be well placed in Nature Comm.

The experimental approaches are technically sound and results well and appropriately reported.

I have just a few very minor comments for improving this excellent article.

1.

Page 15/16 wrong Figure references, the MST data is shown in Figure 4E; TbCBP66 colP in Figure 4F

Line 334-337

The affinity of a TAMRA-labelled TbCBP30 peptide comprising residues R34-G60 towards the TbCBP20-TbCBP110 dimer was determined to $980 \text{ nM} \pm 386 \text{ nM}$, by microscale thermophoresis (Figure 4E, S11).

Line 366-367

Our data reveal that a region within the C-terminus of TbCBP30 (residues K150 to A200) is required for the co-purification of full-length TbCBP66 (Figure 4F).

The figure citations were corrected.

2.

Line 473-475

These observations indicate high stability of the complex and would support that tetrameric CBC forms a permanent assembly.

"And" inserted.

3.

Line 483-485

The RNA processing machinery of *T. brucei* is the target of the drug acoziborole (SCYX-7158), which is effective against human African trypanosomiasis (HAT), demonstrating that drugging the RNA metabolism of trypanosomes is an effective strategy.

Typos corrected.

4.

Line 521

The abbreviation for tris(2-carboxyethyl)phosphine should be introduced and changed to the common 'TCEP'

Same for PMSF

Corrected as requested.

Reviewer #3 (Remarks to the Author):

Bernhard et al report a cryo-EM structural analysis of the *Trypanosoma brucei* cap binding complex (CBC) that is composed of 4 proteins, CBP20/110/30/66, bound to the m7G cap and/or SL RNA. In their cryo-EM focused study, the authors tried to resolve the dimer/trimer/tetramer structures of these proteins to overcome the partial/localized structural ambiguity and understand structural mechanism in the assembly of cap-binding complex. Due to the flexibility in the region of CBP66 and partial CBP30, the authors could not obtain high resolution structures for these two proteins in the complex. SAXS analysis supports the presence of disordered region and the associations of these proteins to the core CBP20/110 complex, where the CBP30 tethers CBP66, like a 'bridge', is supported by mass photometry and size exclusion. By investigating three oligomeric forms, key contact regions are identified, especially CBP30 and 66 to the core complex, and further supported by mutagenesis, MST, CoIP, SEC and EMSA. The authors also confirmed that CBP110 resembles human CBP80, based on structural homology. They recognition of m7GMP in the RNA 5'-cap by CBP20 is revealed. Nevertheless, the final structure remains undefined for partial CBP30 and the whole CBP66.

Overall, this is interesting and relevant work, that provides significant novel insight into an important and kinetoplastid-specific aspects of gene expression, which may be exploited for therapeutic approaches eventually. The data are technically sound, and structural findings are combined with biochemical and biophysical methods. What is missing is to link the structural data with biological functional, e.g. confirming the functional significance of the observed interactions in cellular assays. This would significantly strengthen the impact of the study, but would of course also require significant experimental work .

The authors should consider the following points in a revised manuscript:

Specific comments:

1.- Could the authors support the functional significance for some of the kinetoplastic unique structural features by cellular experiments?

The *in vivo* study of Li and Tschudi 2005 identifies the trypanosome TbCBC subunits CBP20, CBP30 and CBP110 as essential for trypanosome survival with a role in trans-splicing. Due to the essentiality of the subunits, *in vivo* experiments with conditional mutants are resource intensive, i.e. months to generate validated cell lines. We feel that the claims of our study are supported by our biochemical experiments.

2.- It is unfortunate that important structural details are not resolved due to flexibility and thus their contribution to RNA binding are not well resolved. Have the authors considered to provide additional (structural) information for the RNA recognition by RRM and Zn fingers from crystallography or NMR?

As structural biologists, the authors have made multiple attempts to resolve the structure of CBP66 bound to RNA: co-crystallization of different CBP66* constructs with the stem-loop, cryo-EM of CBP66* with stem-loop RNA and the SL RNA exon, cryo-EM of the CBC tetramer with the SL RNA exon, but none of these attempts resulted in meaningful data. To consolidate the role of CBP66 in RNA binding, we have conducted an additional set of mutant experiments and **added this data as Figure S12**.

3.- EMSA was used to characterize SL RNA binding with and w/o cap (Fig. 3 F, G) the data must be shown at least in a supplementary material and fitted parameters including Hill factors provided in a table.

The original uncropped EMSA gels and recorded intensities will be provided as source files for each figure in the final submission. The hill slopes have been added to the supplementary tables.

4.- Fig. 5E: what are the Hill factors, can they be rationalized with the structure?

We have added the Hill factors to Table S2 and S3. A trend for a Hill coefficient >1 suggesting cooperative binding can be observed for the tetramer that contains the two binding sites. Biologically this makes sense: once the SL RNA is tethered to one site in the complex, binding of the second site is favoured. However, accounting for the limited accuracy of EMSA quantification (Figure 5E are FP experiments), the deduction of kinetic parameters may over-interpret the data. With the limited availability of the synthetic cap4 ligand, more detailed analyses are currently not feasible.

5.- Why not using a more precise method to quantify the affinity, e.g. ITC, or fluorescence spectroscopy?

The custom synthesis of cap4-RNA is resource-intensive (see Material and Methods). We settled for label-free EMSA as a label-free method with low material consumption. Other methods would require labelling (fluorescence) or high amounts of materials (ITC). *TbCBC* tends to precipitate at concentrations >2 mg/ml.